# Node-Level Differentially Private Graph Neural Networks

## Abstract

Graph Neural Networks (GNNs) are a popular technique for modelling graph-structured data that compute node-level representations via aggregation of information from the local neighborhood of each node. However, this aggregation implies increased risk of revealing sensitive information, as a node can participate in the inference for multiple nodes. This implies that standard privacy preserving machine learning techniques, such as differentially private stochastic gradient descent (DP-SGD) – which are designed for situations where each data point participates in the inference for one point only – either do not apply, or lead to inaccurate solutions. In this work, we formally define the problem of learning 1-layer GNNs with node-level privacy, and provide an algorithmic solution with a strong differential privacy guarantee. Even though each node can be involved in the inference for multiple nodes, by employing a careful sensitivity analysis and a non-trivial extension of the privacy-by-amplification technique, our method is able to provide accurate solutions with solid privacy parameters. Empirical evaluation on standard benchmarks demonstrates that our method is indeed able to learn accurate privacy preserving GNNs, while still outperforming standard non-private methods that completely ignore graph information.

## 1 Introduction

Graph Neural Networks (GNNs) are powerful modeling tools that capture structural information provided by a graph. Consequently, they have become popular in a wide array of domains such as biology (Ktena et al., 2018), medicine (Ahmedt-Aristizabal et al., 2021), chemistry (McCloskey et al., 2019), computer vision (Wang et al., 2019), and text classification (Yao et al., 2019).

GNNs allow aggregation of data from the neighbors of a given node in the graph, thus evading the challenge of data scarcity per node. Naturally, such solutions are quite attractive in modeling users – each node of the graph is represented by the user and the connections represent interactions between the users – for a variety of recommendation/ranking tasks, where it is challenging to obtain and store user data (Fan et al., 2019; Budhiraja et al., 2020; Levy et al., 2021).

However, such solutions are challenging to deploy as they are susceptible to leaking highly sensitive private information about the users. It is well-known that standard ML models – without GNN style data aggregation – can leak highly sensitive information about the training data (Carlini et al., 2019). The risk of leakage is significantly higher in GNNs as each prediction is based on not just the individual node, but also an aggregation of data from the neighborhood of the given node. In fact, there are two types of highly-sensitive information about an individual node that can be leaked: a) the features associated with each node/user, b) the connectivity information of an individual node/user.

In this work, we study the problem of designing algorithms to learn GNNs while preserving *node-level* privacy, i.e., preserving both the features as well as connectivity information of an individual node. We use differential privacy as the notion of privacy (Dwork et al., 2006) of a node, which roughly-speaking requires that the algorithm should learn similar GNNs despite perturbation of an entire node and *all* the data points or predictions associated with that node.

Example scenarios for such a solution include ranking/recommendation of entities like documents/emails in an organization. Here, the graph can be formed by a variety of means like how users interact with each other, and the goal would be to learn user features that can enable more

accurate ranking of emails/documents. Naturally, user interaction data as well as individual users' features (like the topics in which user is interested in) would be critical to preserve, and any revelation of such data can be catastrophic. Furthermore, once GNNs are learned to model users while preserving privacy, they can be used in different settings based on the problem requirement. For example, in settings where a node can access it's $r$-hop neighbors data, we can directly apply $r$-layer GNNs (if they are trained with DP). Similarly, in certain scenarios, we would want to learn GNNs over a large enterprise and deploy the same model for a small enterprise, where at inference time neighborhood information (like managerial reporting structure) might be publicly accessible within the enterprise but not across enterprises. See Section 4 for a detailed discussion.

Recent works have explored the problem of differentially private learning of GNNs, but they either consider a restricted setting of edge-level privacy which is often insufficient for real-world problems or they restrict themselves to simpler settings like bipartite graphs or node-level privacy without preserving individual connectivity information (Wu et al., 2021a;b; Zhou et al., 2020).

In contrast, our proposed method preserves the privacy of the features of each node ('user'), their labels as well as their connectivity information. To this end, we adapt the standard DP-SGD method (Song et al., 2013; Bassily et al., 2014; Abadi et al., 2016) to our setting. But, analysis of the standard DP-SGD method does not directly extend to GNNs, as each *gradient* term in GNNs can depend on multiple nodes. The key technical contribution of our work is two-fold: i) we provide a careful sensitivity analysis for the special case of 1-layer GNNs, ii) we extend the standard privacy by amplification technique to GNNs where one gradient term can depend on multiple users. Note that the standard privacy by amplification method only applies to scenarios where each point corresponds to one user/entity. By combining the above two results with the standard Rényi Differential Privacy (RDP) accounting, we obtain a formal proof of privacy for our method.

Finally, we evaluate our DP-GNN method on standard benchmarks. We demonstrate that DP-GNN is reasonably accurate compared to the standard 1-layer GCN models, while providing privacy parameters of about $\leq 30$ which are close to the industry standard. More critically, compared to standard MLP (multi-layer perceptron) based methods that completely discard graph side-information, our method can be 5-6% more accurate while still providing strong privacy guarantees. That is, we demonstrate that GNN based techniques can indeed be deployed in practice with the benefits of improved accuracy over vanilla MLP style methods while still preserving sensitive user data.

**Contributions:** We propose a Node-Level Differentially Private Graph Neural Network that works well in practice and provides formal privacy guarantees. This is the first work, to the best of our knowledge, to provide such strong privacy guarantees for each individual node in the graph learning regime. Our main contributions are organised as follows:

- Formulation: In Section 3, we formalize the problem of node-level differentially private GNNs, and discuss various important settings in which a solution to the problem is applicable.

- Method: In Section 4, we describe our algorithm that adapts standard DP-SGD to train differentially private GNNs, with a strong privacy guarantee that extends standard privacy amplification by sampling.

- Empirical Evaluation: In Section 5, we evaluate our framework on multiple benchmark graph datasets on the task of node classification. We demonstrate that our DP-GNN method can outperform non-private and private MLP methods that cannot utilize graph information.

## 2 RELATED WORK

Mechanisms to make the training process of machine learning models private primarily fall into two categories: model-agnostic methods such as PATE (Papernot et al., 2017), and model-aware methods such as DP-SGD (Abadi et al., 2016), which augment the standard paradigm of gradient-based training to be differentially private. DP-SGD, in particular, has been used successfully to train neural network models to classify images (Abadi et al., 2016) and text (Anil et al., 2021).

Today, there are many varieties of graph neural networks employed: Graph Convolutional Neural Networks (Kipf & Welling, 2016), Graph Attention Networks (Veličković et al., 2018), GraphSAGE (Hamilton et al., 2017), and Message-Passing Neural Networks (Gilmer et al., 2017), to name a few. Broadly, these models compute node-level representations via aggregation of neighbourhood-level

information, that can lead to diffusion of private information across multiple nodes, thus making application of standard DP-SGD like techniques non-trivial.

There has been recent work in learning and evaluating edge-level private GNNs (Wu et al., 2021b) but they do not preserve node-level data. Private GNNs have also been studied from the perspective of local privacy (Sajadmanesh & Gatica-Perez, 2020), where each node performs its share of the GNN computation locally. In such a setting, each node sends noisy versions of its features and labels to neighbouring nodes in order to learn shared weights, resulting in a elaborate learning algorithm that needs to correct for the bias in both the features and labels. (Wu et al., 2021a) utilizes private GNNs for recommendation systems, but their method assumes a bipartite graph structure, and cannot naturally handle homogeneous graphs. Other approaches employ federated learning (Zhou et al., 2020), but only guarantee that the GNN neighbourhood aggregation step is differentially private, which is insufficient to guarantee privacy of each node's neighborhood. Finally, other attempts (Shan et al., 2021) to create privacy-preserving GNNs exist, but these do not use the formal notion of DP.

Model-agnostic methods, such as PATE, have recently been investigated to train GNNs (Olatunji et al., 2021). In their current form, however, such methods require access to public data samples, which may not always be available for the task at hand.

In contrast to previous approaches which protect the privacy of a node's features and labels only, we additionally seek to protect every node's *adjacency vector*, which is its private list of connections to neighbouring nodes. This is because the existence of communication between a pair of nodes can often be sensitive information in itself. Further, our approach extends the standard approaches of gradient-based training to scalably train node-level differentially private GNNs in a *centralized* setting, without any access to public data. Depending on the required privacy setting, this mechanism can be composed with locally differentially private mechanisms to generate node-level predictions.

In different contexts, there has been extensive work on node-level DP (Raskhodnikova & Smith, 2016; Karwa et al., 2011; Borgs et al., 2015; 2018). But these methods generally deal with modeling 'global' graph-level statistics and do not support learning methods such as GNNs. In contrast, our approach aims to *predict* 'local' node-level statistics (like the label of a node) while preserving node-level privacy.

## 3 PROBLEM FORMULATION AND PRELIMINARIES

Consider a graph dataset $G = (V, E, \mathbf{X}, \mathbf{Y})$ with *directed* graph $\mathcal{G} = (V, E)$ represented by a adjacency matrix $\mathbf{A} \in \{0, 1\}^{n \times n}$. $n$ is the number of nodes in $\mathcal{G}$, $V$ denotes the node set, $E$ denotes the edge set. Each node $v$ in the graph is equipped with a feature vector $\mathbf{X}_v \in \mathbb{R}^d$; $\mathbf{X} \in \mathbb{R}^{n \times d}$ denotes the feature matrix. $\mathbf{Y} \in \mathbb{R}^{n \times Q}$ is the label matrix and $\mathbf{y}_v$ is the label for the $v$-th node over $Q$ classes. Note that many of the labels in the label vector can be missing, which models the semi-supervised setting. In particular, we assume that node labels $\mathbf{y}_v$ are only provided for a subset of nodes $V_{tr} \subset V$, called the training set.

Given the graph dataset $G$, the goal is to learn parameters of a one-layer GNN while preserving privacy of individual nodes. A GNN can be represented by the following operations:

$$\widehat{\mathbf{y}}_v = \mathsf{GNN}(\mathbf{A}, \mathbf{X}, v; \mathbf{\Theta}) := f_{\text{dec}}\left(f_{\text{agg}}\left(\{f_{\text{enc}}(\mathbf{X}_u) \mid \mathbf{A}_{vu} \neq 0\}\right)\right) \tag{1}$$

where $\widehat{\mathbf{y}}_v$ is the prediction from the GNN for a given node $v$, $f_{\text{enc}}$ is the encoder function that encodes node features with parameters $\Theta_{\text{enc}}$, $f_{\text{agg}}$ is the neighborhood aggregation function with parameters $\Theta_{\text{agg}}$, $f_{\text{dec}}$ is the prediction decoder function with parameters $\Theta_{\text{dec}}$, and $\Theta := (\Theta_{\text{enc}}, \Theta_{\text{agg}}, \Theta_{\text{dec}})$.

While our results apply to most 1-layer GNN models (Hamilton et al., 2017; Veličković et al., 2018; Xu et al., 2018), for simplicity, we focus on 1-layer Graph Convolutional Network (GCN) models[1] (Kipf & Welling, 2016). These GCN models use a multi-layer perceptron (MLP) for encoder and decoder functions, with non-linear activation function $\sigma$:

$$\widehat{\mathbf{y}}_v = \mathsf{GCN}(\mathbf{A}, \mathbf{X}, v; \mathbf{\Theta}) := \mathsf{MLP}_{\text{dec}}\left(\mathbf{A}_v \sigma(\mathsf{MLP}_{\text{enc}}(\mathbf{X}))\Theta_{\text{agg}}\right) \tag{2}$$

---

[1] As is common in practice, we allow any normalization and addition of self-loops to $\mathbf{A}$.

Thus, "learning" a GCN is equivalent to finding parameters $\Theta := (\Theta_{\text{enc}}, \Theta_{\text{agg}}, \Theta_{\text{dec}})$ that minimize a suitable loss:

$$\Theta^* = \arg\min_{\Theta} \underbrace{\sum_{v \in V} \ell(\hat{\mathbf{y}}_v; \mathbf{y}_v)}_{\mathcal{L}(G, \Theta)} \tag{3}$$

where $\ell : \mathbb{R}^{Q \times Q} \to \mathbb{R}$ is a standard loss function such as categorical cross-entropy.[2]

As mentioned earlier, we use differential privacy as the notion of privacy of a node. Before defining differential privacy, we first define the notion of adjacent graph datasets:

**Definition 1** (Adjacent Graph Datasets). *Two graph datasets $G$ and $G'$ are said to be **node-level adjacent** if one can be obtained by adding or removing a node (with its features, labels and associated edges) to the other. That is, $G$ and $G'$ are exactly the same except for the $v$-th node, i.e., $\mathbf{X}_v$, $\mathbf{y}_v$ and $\mathbf{A}_v$ differ in the two datasets.*

Informally, $\mathcal{A}$ is said to be node-level differentially-private algorithm if the addition or removal of a node in $\mathcal{A}$'s input does not affect $\mathcal{A}$'s output significantly.

**Definition 2** (Node-level Differential Privacy). *Consider any randomized algorithm $\mathcal{A}$ that takes as input a graph dataset. $\mathcal{A}$ is said to be $(\alpha, \gamma)$ **node-level Rényi differentially-private** (Mironov, 2017b) if, for every pair of node-level adjacent datasets $G$ and $G'$:*

$$D_{\alpha}(\mathcal{A}(G) \parallel \mathcal{A}(G')) \leq \gamma,$$

*where **Rényi divergence** $D_{\alpha}$ of order $\alpha$ between two random variables $P$ and $Q$ is defined as:*

$$D_{\alpha}(P \parallel Q) = \frac{1}{\alpha - 1} \ln \mathbb{E}_{x \sim Q} \left[ \frac{P(x)}{Q(x)} \right]^{\alpha}.$$

Note that we use Rényi differentially-private (RDP) (Mironov, 2017b) as the formal notion of differential privacy (DP), as it allows for tighter composition of DP across multiple steps. This notion is closely related to the standard $(\varepsilon, \delta)$-differential privacy (Dwork et al., 2006); Proposition 3 of Mironov (2017b) states that any $(\alpha, \gamma)$-RDP mechanism also satisfies $(\gamma + \frac{\log 1/\delta}{\alpha - 1}, \delta)$-differential privacy for any $0 < \delta < 1$.

Thus, the goal is to find $\Theta$ by optimizing equation 3 while ensuring RDP (Definition 2). It is clear that node-level privacy is essential when training models on graph datasets with sensitive node-level information. However, node-level privacy is significantly harder to achieve than the weaker notion of edge-level privacy. In the context of GNNs, the representation for a node is computed using not just the node's individual features, but also features of other nodes from the local neighbourhood. Thus, the removal of a node from a graph dataset affects its entire local neighbourhood, which can be a very large set of nodes. This is in contrast to the standard non-graph setting for differentially private models, where the representation of individual users would only depend on the user's own data.

We now define two concepts that are critical in our design and analysis of a private GNN learning method.

**Definition 3.** *The **node-level sensitivity** $\Delta(f)$ of a function $f$ defined on graph datasets is:*

$$\Delta(f) = \max_{\substack{\text{node-level adjacent} \\ G, G'}} \|f(G) - f(G')\|_2$$

*The $K$-**restricted node-level sensitivity** $\Delta_K(f)$ of a function $f$ defined on graph datasets is:*

$$\Delta_K(f) = \max_{\substack{\deg(G), \, \deg(G') \leq K \\ \text{node-level adjacent} \\ G, G'}} \|f(G) - f(G')\|_2$$

**Definition 4.** *We define the **clipping** operator $\text{Clip}_C(.)$ as:* $\text{Clip}_C(v) = \min\left(1, \frac{C}{\|v\|_F}\right) \cdot v$, *for any vector or matrix $v$.*

---

[2] The analysis here holds for multi-label settings as well, which would instead use loss functions such as sigmoidal cross-entropy, for example.

---

**Algorithm 1:** DP-GNN (SGD): Differentially Private Graph Neural Network with SGD

---

**Data:** Graph $G = (V, E, \mathbf{X}, \mathbf{Y})$, GNN definition GNN, Training set $V_{tr}$, Loss function $\mathcal{L}$,
      Batch size $m$, Maximum degree $K$, Learning rate $\eta$, Clipping threshold $C$, Noise
      standard deviation $\sigma$, Maximum training iterations $T$.

**Result:** GNN parameters $\mathbf{\Theta}_T$.

Note that $V_{tr}$ is the subset of nodes for which labels are available (see Paragraph 1 of Section 3).

Using $V_{tr}$, construct the set of training subgraphs $S_{tr}$ with Algorithm 2.

Construct the $0-1$ adjacency matrix $\mathbf{A}$: $\mathbf{A}_{vu} = 1 \iff (v, u) \in S_{tr}$

Initialize $\mathbf{\Theta}_0$ randomly.

**for** $t = 0$ **to** $T$ **do**
    | Sample set $\mathcal{B}_t \subseteq V_{tr}$ of size $m$ uniformly at random from all subsets of $V_{tr}$.
    | Compute the update term $\mathbf{u}_t$ as the sum of the clipped gradient terms in the batch $\mathcal{B}_t$:

$$\mathbf{u}_t \leftarrow \sum_{v \in \mathcal{B}_t} \mathrm{Clip}_C(\nabla_{\mathbf{\Theta}} \ell\left(\mathsf{GNN}(\mathbf{A}, \mathbf{X}, v; \mathbf{\Theta}_t); \mathbf{y}_v\right))$$

    | Add independent Gaussian noise to the update term: $\tilde{\mathbf{u}}_t \leftarrow \mathbf{u}_t + \mathcal{N}(0, \sigma^2 \mathbb{I})$
    | Update the current estimate of the parameters with the noisy update: $\quad \mathbf{\Theta}_{t+1} \leftarrow \mathbf{\Theta}_t - \frac{\eta}{m} \tilde{\mathbf{u}}_t$

**end**

---

# 4    LEARNING GRAPH CONVOLUTIONAL NETWORKS (GCN) VIA DP-SGD

In this section, we provide a variant of DP-SGD (Bassily et al., 2014) designed specifically for GCNs (Equation 2), and show that our method guarantees node-level DP (Definition 2).

The first step in our method is to subsample the neighborhood of each node to ensure that each node has only $K$ neighbors. This is important to ensure that influence of a single node is restricted to only $K$ other nodes. Next, similar to standard mini-batch SGD technique, we sample a subset $\mathcal{B}_t$ of $m$ nodes chosen uniformly at random from the set $V_{tr}$ of training nodes. In contrast to the standard mini-batch SGD, that samples points with replacement for constructing a mini-batch, our method samples mini-batch $\mathcal{B}_t$ uniformly from the set of all training nodes. This distinction is important for our privacy amplification result. Once we sample the mini-batch, we apply the standard DP-SGD procedure of computing the gradient over the mini-batch, clipping the gradient and adding noise to it, and then use the noisy gradients for updating the parameters.

DP-SGD requires each update to be differentially private. In standard settings where each gradient term in the mini-batch corresponds to only one point, we only need to add $O(C)$ noise – where $C$ is the clipping norm of the gradient – to ensure privacy. However, in the case of GCNs with node-level privacy, perturbing one node/point $\widehat{\mathbf{v}}$ can have impact loss term corresponding to all its neighbors $\mathcal{N}_{\widehat{\mathbf{v}}}$. So, to ensure the privacy of each update, we add noise according to the sensitivity of aggregated gradient $\nabla_{\mathbf{\Theta}} \mathcal{L}(\mathcal{B}_t; \mathbf{\Theta}_t) := \sum_{v \in \mathcal{B}_t} \mathrm{Clip}_C(\nabla_{\mathbf{\Theta}} \ell\left(\mathsf{GCN}(\mathbf{A}, \mathbf{X}, v; \mathbf{\Theta}_t); \mathbf{y}_v\right))$ wrt an individual node $\widehat{\mathbf{v}}$. To this end, we provide a finer bound in Lemma 1 on the sensitivity of $\nabla_{\mathbf{\Theta}} \mathcal{L}(\mathcal{B}_t; \mathbf{\Theta}_t)$ based on the maximum degree of the graph $G$.

In traditional DP-SGD, a crucial component in getting a better privacy/utility trade-off over just adding noise according to the sensitivity of the minibatch gradient, is privacy amplification by sampling (Kasiviswanathan et al., 2008; Bassily et al., 2014). This says that if an algorithm $\mathcal{A}$ is $\varepsilon$-DP on a data set $D_1$, then on a random subset $D_2 \subseteq D_1$ it satisfies roughly $\frac{|D_2|}{|D_1|} (e^\varepsilon - 1)$-DP. Unlike traditional ERMs, we cannot directly use this result in the context of GCNs. The reason is again that on two adjacent data sets, multiple loss terms corresponding to $\widehat{\mathbf{v}}$ and its neighbors $\mathcal{N}_{\widehat{\mathbf{v}}}$ get modified. To complicate things further, the minibatch $\mathcal{B}_t$ that gets selected may only contain a small random subset of $\mathcal{N}_{\widehat{\mathbf{v}}}$. To address these issues, we provide a new privacy amplification theorem (Theorem 1). To prove the theorem, we adapt (Feldman et al., 2018, Lemma 25) – that shows a weak form of convexity of Renyi divergence – for our specific instance, and provide a tighter bound by exploiting the special structure in our setting along with the bound on sensitivity discussed above.

**Theorem 1** (Amplified Privacy Guarantee for any 1-Layer GCN)**.** *Consider the loss function* $\mathcal{L}$ *of the form:* $\mathcal{L}(G, \mathbf{\Theta}) = \sum_{v \in V_{tr}} \ell\left(\mathsf{GCN}(\mathbf{A}, \mathbf{X}, v; \mathbf{\Theta}_t); \mathbf{y}_v\right).$ *Recall,* $N$ *is the number of training nodes* $V_{tr}$, $K$ *is an upper bound on the maximum degree of the input graph, and* $m$ *is the batch size.*

*For any choice of the noise standard deviation $\sigma > 0$ and clipping threshold $C$, every iteration $t$ of Algorithm 1 is $(\alpha, \gamma)$ node-level Rényi DP, where:*

$$\gamma = \frac{1}{\alpha - 1} \ln \mathbb{E}_\rho \left[ \exp \left( \alpha(\alpha - 1) \cdot \frac{2\rho^2 C^2}{\sigma^2} \right) \right], \quad \rho \sim \text{Hypergeometric}(N, K + 1, m).$$

Hypergeometric *denotes the standard hypergeometric distribution (Forbes et al., 2011).*

*By the standard composition theorem for Rényi Differential Privacy (Mironov, 2017b), over $T$ iterations, Algorithm 1 is $(\alpha, \gamma T)$ node-level Rényi DP, where $\gamma$ and $\alpha$ are defined above.*

See Appendix A for a detailed proof.

**Remark 1**: Roughly, for $m \gg K$ and for $T = O(1)$, the above bound implies $\sigma = O(K)$ noise to be added per step to ensure RDP with $\alpha = O(1)$ and $\gamma = O(1)$. In contrast, the standard DP-SGD style privacy amplification do not apply to our setting as each gradient term can be impacted by multiple nodes.

**Remark 2**: We provide node-level privacy, that is the method preserves neighborhood information of each node as well. But, we require asymmetric/directed graph, that is, changing a row in the adjacency matrix does not impact any other part of the matrix. This is a natural assumption in a variety of settings, for example, in social networks when the graph is constructed by "viewership" data, edge $(v, v')$ exists iff user $v$ viewed a post from user $v'$.

**Remark 3**: While we provide a formal privacy guarantee for 1-layer GCNs, the same applies for any 1-layer GNN model.

**Remark 4**: We adapt a DP version of the Adam (Kingma & Ba, 2014; TFP) optimizer to the GNN setting, called DP-GNN (Adam), with details in Appendix D.

**Privacy at Inference Time**: Note that Theorem 1 guarantees that the GCN parameters $\Theta$ that are learnt via Algorithm 1 preserve privacy. However, unlike standard ML models where prediction for each point depends only on the model parameters $\Theta$ and the point itself, the privacy of $\Theta$ does not imply that inference using the GCN model (or any GNN model) will be privacy preserving. In general, the inference about node $v$ can reveal information about its neighbors $\mathcal{N}_v$. Broadly, there are three settings where we can infer labels for a given node while preserving privacy:

1. Each node has access to the features of its neighbors. In this setting, the aggregation of features from the neighbors does not lead to any privacy loss. Several real-world problems admit such a setting: for example, in social networks where any user has access to a variety of activities/documents/photos of their friends (neighbors).

2. Node features are completely private. In this setting, a node $v$ does not have direct access to the features of its neighbors $\mathcal{N}_v$. Here, the standard GCN model is not directly applicable, but we can still apply GCNs by aggregating the neighborhood features with *noise*. Generally, the resulting prediction for a node would be meaningful only if the degree of the node is reasonably large.

3. Training and test graph datasets are disjoint. In this setting, the goal is to privately learn $\Theta$ using the training graph, that can be 'transferred' to the test graphs. Additionally, the feature information is shared publicly within test graph dataset nodes. A variety of problems can be modeled by this setting: organizations can be represented by a graph over its employees, with the goal to learn a private ranking/recommendation model that can easily be adapted for completely distinct organizations.

While there are multiple problems that can be modeled by the above mentioned settings, we focus on the first setting for our empirical results.

## 5 EXPERIMENTAL RESULTS

In this section, we present empirical evaluation of our method on standard benchmarks from the widely used Open Graph Benchmark (OGB) suite (Hu et al., 2020). The goal is to demonstrate that our method (DP-GNN) can indeed learn privacy preserving 1-layer GCNs accurately.

Table 1: **Test accuracy of DP-GNN compared to the baselines on different datasets.** DP-GNN clearly performs better than the Private and Non-Private MLP baselines across these datasets.

| Algorithm | ogbn-arxiv | ogbn-products | ogbn-mag |
|---|---|---|---|
| GCN | $68.422 \pm 0.267$ | $76.139 \pm 0.519$ | $34.680 \pm 0.424$ |
| **DP-GNN** (Adam) | $63.934 \pm 0.469$ | $69.576 \pm 0.276$ | $30.059 \pm 0.252$ |
| **DP-GNN** (SGD) | $64.137 \pm 0.621$ | $69.041 \pm 0.126$ | $30.147 \pm 0.241$ |
| MLP | $55.236 \pm 0.317$ | $61.364 \pm 0.132$ | $26.969 \pm 0.361$ |
| DP-MLP | $53.462 \pm 0.242$ | $61.064 \pm 0.110$ | $25.259 \pm 0.321$ |

As mentioned earlier, in several data critical scenarios, practitioners cannot use sensitive graph information, and have to completely discard GNN based models due to *privacy concerns*. Hence, the main benchmark of our evaluation is to demonstrate that DP-GNN is able to provide *more accurate solutions* than standard methods that completely discard the graph information. The key baselines for our method are both standard non-private MLP models as well as differentially private MLP models trained using DP-SGD and DP-Adam. We also compare against the standard 1-layer GCNs (without any privacy guarantees) as it bounds the maximum accuracy we can hope to achieve out of our method.

## 5.1 DATASETS AND SETUP

**OGB datasets:** We use three moderate-to-large sized node classification datasets from OGB suite[3]: ogbn-arxiv, ogbn-products and ogbn-mag. The ogbn-arxiv and ogbn-mag datasets consist of papers extracted from the Microsoft Academic Graph (MAG) dataset (Wang et al., 2020). The ogbn-arxiv dataset is a paper citation network of arxiv papers and consists of around 169K nodes, while the ogbn-mag dataset is a heterogenous graph with node types papers, authors, institutions and topics and consists of around 1.9M nodes. However, following the standard approach in (Hu et al., 2020) we create a homogeneous graph of papers (736K nodes) from the ogbn-mag dataset. The ogbn-products dataset is an Amazon products co-purchasing network and consists of 2.4M nodes. Each dataset consists of edges, node features and labels (multi-class), and is split into standard train, test and validation sets (Hu et al., 2020). Finally, following (Hu et al., 2020), we consider the transductive semi-supervised setting for all the datasets, i.e., the entire graph is available during training but only a few nodes in $V_{tr}$ have labels available. See Appendix E for additional details about the datasets.

**Gradient Clipping:** For DP-GNN, we perform layer-wise gradient clipping, i.e., the gradients corresponding to the encoder, aggregation and decoder functions are clipped independently with different clipping thresholds. For each layer, the clipping threshold $C$ in Algorithm 1 is chosen as $C_f \times C_\%$ where $C_f$ is a scaling factor and $C_\%$ is the 75th percentile of gradient norms for that layer at initialization on the training data. We finetune the $C_f$ parameter for each dataset. We set the noise for each layer $\sigma$ such that the noise multiplier $\lambda = \frac{\sigma}{2(K+1)C}$ is identical for each layer, where $\sigma/\lambda$ is essentially the sensitivity. It is not hard to observe that the overall privacy cost only depends on $\lambda$.

**Methods:** We benchmark the following methods: a) **DP-GNN**: Our method (Algorithm 1) specialized for a 1-layer GCN with an MLP as the encoder and the decoder, b) **GCN**: A 1-layer GCN with an MLP encoder and decoder. This defines the highest possible numbers for our method but due to privacy concerns, non-private GCN might not be suitable for deployment in practice, c) **MLP**: A standard multi-layer perceptron (MLP) architecture on the raw node features as proposed in prior works (Hu et al., 2020). This model does not utilize any graph level information, d) **DP-MLP**: A DP version of MLP (with standard architecture) trained using DP-Adam (TFP).

**Detailed Setup and Hardware:** DP-GNN and all the aforementioned baselines are implemented in TensorFlow 2.0 (Abadi et al., 2015) using Graph Nets[4] and Sonnet[5]. All experiments are performed on 2x2 TPU v2 Pods. We perform model selection for all the methods based on their performance on the validation set. We run each experiment nine times and report the mean and standard deviation for performance on the test set in Table 1.

**Hyperparameter Tuning:** We perform exhaustive grid search over batch size, learning rate, activation functions, and number of encoder and decoder MLP layers for the non-private baselines.

---

[3] ogb.stanford.edu/docs/nodeprop   [4] github.com/deepmind/graph_nets   [5] github.com/deepmind/sonnet

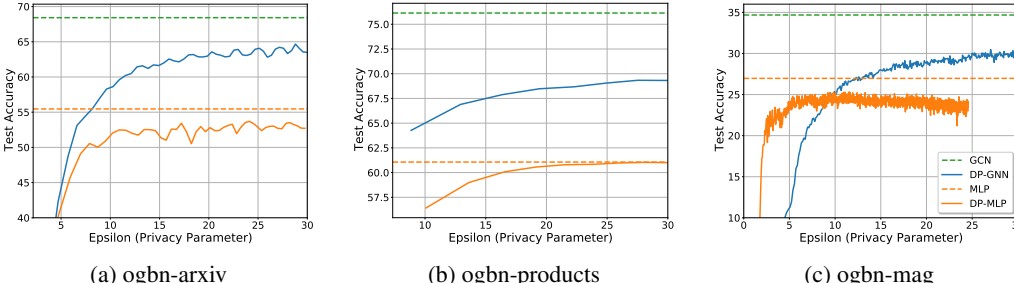

|  |  |  |
|:-:|:-:|:-:|
| (a) ogbn-arxiv | (b) ogbn-products | (c) ogbn-mag |

Figure 1: **Performance of the proposed DP-GNN method as well as the baselines on the ogbn-arxiv, ogbn-products and ogbn-mag datasets.** Clearly, DP-GNN offers a performance better than both of the Non-Private MLP and DP-MLP methods, with a reasonable privacy budget of $\varepsilon \leq 30$.

Additionally, we tune over noise multiplier ($\sigma$ in Algorithm 1) and clipping thresholds for the private baselines. We provide detailed information regarding the hyperparameters in Appendix E.

**Results:** Table 1 compares DP-GNN's accuracy against baselines on the ogbn-arxiv, ogbn-products and ogbn-mag datasets. We extensively tune baselines on the three datasets as mentioned above and are able to replicate, and in some cases, improve the reported performance numbers for the baselines (Hu et al., 2020). We use the higher number of the two for comparison with our method.

Overall, we observe that our proposed method DP-GNN significantly outperforms the Non-Private MLP (without any usage of the graphs) and DP-MLP (trained using standard DP-Adam) baselines on all of the datasets and with a reasonable privacy budget of $\varepsilon \leq 30$. For example, for ogbn-arxiv dataset, our method DP-GNN (SGD) is about 8% more accurate than MLP and 10% more accurate than DP-MLP. Similarly, for ogbn-products our method is about 5% more accurate than both MLP and DP-MLP. Note that we also present numbers for DP-GNN (Adam) (see Appendix D) that uses Adam as the optimizer instead of SGD, as mentioned in Algorithm 1. Also, note that for the rest of the section we use DP-GNN (Adam) for generating accuracy numbers.

Next, Figure 1 provides a comparison of epsilon vs test set accuracy for the three benchmark datasets. Note that for $\varepsilon \geq 10$, DP-GNN is significantly more accurate than DP-MLP. It is interesting to note that for about $\varepsilon \geq 10$, the accuracy of the DP-MLP saturates and does not increase significantly. In contrast, the accuracy of DP-GNN keeps on increasing with larger $\varepsilon$, and is in general much higher than both MLP and DP-MLP for higher values of $\varepsilon$. Finally, on ogbn-products, DP-GNN is about 5% more accurate than DP-MLP for the entire range of considered values for $\varepsilon$, and is about 2% more accurate than MLP for $\varepsilon = 10$.

Typically, for training non-convex learning models with user-level DP, $\varepsilon \leq 10$ has become a popular choice (Papernot et al., 2020; Kairouz et al., 2021). But as the problem is more challenging in the case of GNNs – multiple nodes can affect inference for a given node and we intend to protect privacy at the node-level – higher $\varepsilon$ seems like a reasonable choice to encourage reasonable solutions. Moreover, as we observe on the ogbn-products dataset, larger dataset sizes can ensure better performance for the standard $\varepsilon$ values as well. Also, our algorithms satisfy stronger Rényi DP properties (Mironov, 2017b), which provide additional protection over traditional $(\varepsilon, \delta)$-DP guarantees.

## 5.2 ABLATION STUDIES

**Batch size** $m$: As has been noted in other DP-SGD works (Abadi et al., 2016; Bagdasaryan et al., 2019), we empirically observe that increasing the batch size helps the performance of the learnt DP-GNN, up to a point. There are multiple effects at play here.

Larger batch sizes imply that the effective noise added per DP-SGD update step is smaller. Thus, training is more stable with larger batch sizes, as Figure 2 shows. Furthermore, effective privacy budget ($\varepsilon$) provided by amplification result has a term of the form $\exp(\varepsilon_0) - 1$ where $\varepsilon_0$ is the privacy budget for a step. So, unless $\varepsilon_0$ is small enough, i.e., the batch size is large enough, the amplification result would be weak. On the other hand, larger batch sizes tend to hurt generalization and training speed, even in the non-private case, as the second column of Table 2 shows.

Thus, there is a trade-off between model performance, privacy budget and batch size. As the last column of Table 2 shows, the difference in performance between private and non-private models

Table 2: **GCN and DP-GNN on the ogbn-arxiv dataset with different batch sizes. The privacy budget for DP-GNN is** $\varepsilon \leq 30$.

| Batch Size | GCN ($A_{\text{GCN}}$) | DP-GNN ($A_{\text{DP-GNN}}$) | $A_{\text{GCN}} - A_{\text{DP-GNN}}$ |
|---|---|---|---|
| 100 | 68.075 | 40.814 | 27.261 |
| 500 | 68.393 | 58.882 | 9.511 |
| 1250 | 68.572 | 61.307 | 7.265 |
| 2500 | 68.356 | 63.025 | 5.331 |
| 5000 | 68.490 | 64.345 | 4.145 |
| 10000 | 68.062 | 64.304 | 3.758 |
| 20000 | 68.491 | 62.062 | 6.429 |

Table 3: **GCN and DP-GNN on the ogbn-arxiv dataset with different degrees. The privacy budget for DP-GNN is** $\varepsilon \leq 30$.

| Degree | GCN ($A_{\text{GCN}}$) | DP-GNN ($A_{\text{DP-GNN}}$) | $A_{\text{GCN}} - A_{\text{DP-GNN}}$ |
|---|---|---|---|
| 3 | 68.563 | 63.439 | 5.124 |
| 5 | 69.020 | 63.940 | 5.080 |
| 7 | 68.945 | 64.599 | 4.346 |
| 10 | 68.372 | 64.103 | 4.269 |
| 15 | 68.224 | 63.522 | 4.702 |
| 20 | 68.642 | 63.054 | 5.588 |
| 32 | 68.152 | 61.901 | 6.251 |

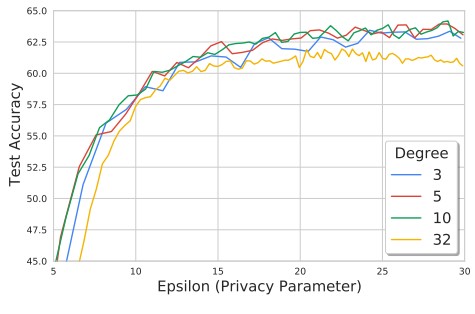

(a) Varying Batch Size $m$      (b) Varying Maximum Degree $K$

Figure 2: **Ablation studies on DP-GNN on the ogbn-arxiv dataset.** (a) shows privacy-utility curves for a range of batch sizes for the DP-GNN. (b) shows privacy-utility curves when varying maximum degree $K$ for the DP-GNN. In both analyses, the other hyperparameters are kept fixed.

tends to diminish as the batch size increases. However, for the reasons pointed out above, beyond a batch size of 10000, the accuracy goes down, as quantified by Table 2.

**Maximum Degree** $K$: Compared to the batch size, the maximum degree $K$ has less of an effect on both non-private and private models trained on ogbn-arxiv, as Table 3 shows. Generally, there is still a trade-off: a smaller $K$ means lesser differentially private noise added at each update step, but also fewer neighbours for each node to aggregate information from.

Finally, we also conduct experiments to understand performance of DP-GNN conditioned on the frequency of a class (how often a class appears in the dataset), with details in Appendix F. On the whole, these experiments suggest that DP-GNN is able to classify data points of "frequent" classes with reasonable accuracy, but struggles with classification accuracy on the data points of "rarer" classes. This observation is in line with previous claims from (Bagdasaryan et al., 2019; Fioretto et al., 2021) that differentially-private models generally perform worse on low-frequency classes, and represent a critical future direction to study.

## 6 CONCLUSIONS AND FUTURE WORK

In this work, we proposed a method to privately learn 1-layer GNN parameters, that outperforms both private and non-private baselines that do not utilize graph information. Our method ensures node-level differential privacy, by a careful combination of sensitivity analysis of the gradients and a privacy amplification result extended to the GNN style settings. We believe that our work is a first step in the direction of designing powerful GNNs while preserving privacy. Promising avenues for future work include learning more general class of GNNs, investigating inference mechanisms mentioned in Section 4 such as different train and test graph datasets, and understanding utility bounds for GNNs with node-level privacy.

## 7 REPRODUCIBILITY STATEMENT

We have taken all efforts to ensure that the results produced in the paper and the submitted material are reproducible, and the methodology is easy to follow. For our theoretical contributions, we have discussed the problem setup and preliminaries in Section 3, provided a detailed algorithm for our proposed methodology in Section 4 for a sound theoretical understanding of the problem

and our solution. For our empirical results, we have detailed the information needed to reproduce the empirical results in Section 5 of the main paper and Appendix E. We supply all the required information regarding the datasets, their pre-processing and source, implementation details for our method and the baselines, specifics regarding the architectures, hyperparameter search spaces and the best hyperparameters corresponding to our experiments. We are working towards an open source implementation, in the spirit of reproducible research.

## 8 ETHICS STATEMENT

The interest in differentially-private models largely stems from a need to protect the privacy of data samples used to train these models. While we have proposed a mechanism here to learn GNNs in privacy-preserving manner, differential privacy seems to exacerbate existing fairness issues on underrepresented classes as Appendix F indicates. This is a concern across all models trained with differential privacy (Bagdasaryan et al., 2019) that needs to be addressed before such models can be deployed in the real world. While there have been recent attempts (Jagielski et al., 2018; Fioretto et al., 2021) to mitigate the disparate effect of differentially private training, there is still a need for an effective practical solution. We anticipate no other negative consequences of our work.

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

## APPENDIX

This appendix is segmented into four key parts:

1. **Section A** gives detailed proofs for the Lemmas discussed in the main paper and additional lemmas that could not be included in the paper due to space constraints.

2. **Section D** provides a detailed description of learning Graph Convolutional Networks via DP-Adam.

3. **Section E** discusses experimental details for reproducing the results in the main paper.

4. **Section F** provides additional results for analysing the performance of the DP-GNN model as compared to the GCN baseline.

## A  LEMMAS AND PROOFS

**Lemma 1** (Node-Level Sensitivity of any 1-Layer GCN). *Consider the loss function $\mathcal{L}$ of the form:*

$$\mathcal{L}(G, \mathbf{\Theta}) = \sum_{\mathbf{v} \in \mathbf{V}} \ell\left(\mathsf{GCN}(\mathbf{A}, \mathbf{X}, \mathbf{v}; \mathbf{\Theta}); \mathbf{y_v}\right).$$

*Let $\mathcal{B}_t$ be any choice of $m$ unique nodes from a graph $G$ with maximum degree bounded above by $K$. Consider the following quantity $\mathbf{u}_t$ from Algorithm 1:*

$$\mathbf{u}_t(G) = \sum_{v \in \mathcal{B}_t} \mathrm{Clip}_C(\nabla_{\mathbf{\Theta}} \ell\left(\mathsf{GCN}(\mathbf{A}, \mathbf{X}, v; \mathbf{\Theta}_t); \mathbf{y}_v\right))$$

*Note that $\mathbf{u}_t(G)$ is a 'clipped' version of $\nabla_{\mathbf{\Theta}} \mathcal{L}(\mathcal{B}_t; \mathbf{\Theta}_t, G)$:*

$$\nabla_{\mathbf{\Theta}} \mathcal{L}(\mathcal{B}_t; \mathbf{\Theta_t}, \mathbf{G}) = \sum_{v \in \mathcal{B}_t} \nabla_{\mathbf{\Theta}} \ell\left(\mathsf{GCN}(\mathbf{A}, \mathbf{X}, v; \mathbf{\Theta}_t); \mathbf{y}_v\right)$$

*Then, the following inequality holds:*

$$\Delta_K(\mathbf{u}_t) < 2(K + 1)C.$$

*Proof.* Let $G$ be an arbitrary graph dataset with adjacency matrix $\mathbf{A}$ and maximum degree bounded above by $K$. Consider an adjacent graph dataset $G'$ with adjacency matrix $\mathbf{A}'$ formed by removing a single node $\widehat{\mathbf{v}}$ from $G$. We wish to bound the following quantity:

$$\|\mathbf{u}_t(G) - \mathbf{u}_t(G')\|_F$$

For convenience, for any node $v$, we denote the corresponding loss terms $\ell_v$ and $\ell'_v$ as:

$$\ell_v = \ell\left(\mathsf{GCN}(\mathbf{A}, \mathbf{X}, v; \mathbf{\Theta}_t); \mathbf{y}_v\right)$$
$$\ell'_v = \ell\left(\mathsf{GCN}(\mathbf{A}', \mathbf{X}', v; \mathbf{\Theta}_t); \mathbf{y}_v\right)$$

From the definition of $\ell_v$, it is clear that the only gradient terms $\nabla_{\mathbf{\Theta}} \ell_v$ affected when adding or removing node $\widehat{\mathbf{v}}$, are those of its neighbors and $\widehat{\mathbf{v}}$ itself. Thus,

$$\mathbf{u}_t(G) - \mathbf{u}_t(G')$$
$$= \mathrm{Clip}_C(\nabla_{\mathbf{\Theta}} \ell_{\widehat{\mathbf{v}}}) \cdot \mathcal{I}[\widehat{\mathbf{v}} \in \mathcal{B}_t] + \sum_{u \in \mathcal{N}_{\widehat{\mathbf{v}}}} \left(\mathrm{Clip}_C(\nabla_{\mathbf{\Theta}} \ell_u) - \mathrm{Clip}_C(\nabla_{\mathbf{\Theta}} \ell'_u)\right) \cdot \mathcal{I}[u \in \mathcal{B}_t]$$

where $\mathcal{I}$ is the indicator random variable. Taking norms:

$$\|\mathbf{u}_t(G) - \mathbf{u}_t(G')\|_F$$

$$= \left\| \text{Clip}_C(\nabla_{\boldsymbol{\Theta}} \ell_{\widehat{\mathbf{v}}}) \cdot \mathcal{I}[\widehat{\mathbf{v}} \in \mathcal{B}_t] + \sum_{u \in \mathcal{N}_{\widehat{\mathbf{v}}}} (\text{Clip}_C(\nabla_{\boldsymbol{\Theta}} \ell_u) - \text{Clip}_C(\nabla_{\boldsymbol{\Theta}} \ell'_u)) \cdot \mathcal{I}[u \in \mathcal{B}_t] \right\|_F$$

$$\leq \|\text{Clip}_C(\nabla_{\boldsymbol{\Theta}} \ell_{\widehat{\mathbf{v}}}) \cdot \mathcal{I}[\widehat{\mathbf{v}} \in \mathcal{B}_t]\|_F + \sum_{u \in \mathcal{N}_{\widehat{\mathbf{v}}}} \|(\text{Clip}_C(\nabla_{\boldsymbol{\Theta}} \ell_u) - \text{Clip}_C(\nabla_{\boldsymbol{\Theta}} \ell'_u)) \cdot \mathcal{I}[u \in \mathcal{B}_t]\|_F$$

$$\text{(triangle inequality)}$$

$$\leq \|\text{Clip}_C(\nabla_{\boldsymbol{\Theta}} \ell_{\widehat{\mathbf{v}}})\|_F + \sum_{u \in \mathcal{N}_{\widehat{\mathbf{v}}}} \|(\text{Clip}_C(\nabla_{\boldsymbol{\Theta}} \ell_u) - \text{Clip}_C(\nabla_{\boldsymbol{\Theta}} \ell'_u))\|_F$$

$$(\mathcal{I} \in \{0, 1\})$$

$$\leq \|\text{Clip}_C(\nabla_{\boldsymbol{\Theta}} \ell_{\widehat{\mathbf{v}}})\|_F + \sum_{u \in \mathcal{N}_{\widehat{\mathbf{v}}}} (\|\text{Clip}_C(\nabla_{\boldsymbol{\Theta}} \ell_u)\|_F + \|\text{Clip}_C(\nabla_{\boldsymbol{\Theta}} \ell'_u)\|_F)$$

$$\text{(triangle inequality)}$$

$$\leq C + \sum_{u \in \mathcal{N}_{\widehat{\mathbf{v}}}} (C + C)$$

$$\text{(gradient clipping)}$$

$$= C + d_{\widehat{\mathbf{v}}} \cdot 2C$$

$$\text{(definition of } d_{\widehat{\mathbf{v}}})$$

$$= C(2d_{\widehat{\mathbf{v}}} + 1) \leq C(2K + 1) < 2(K + 1)C.$$

$$(d_{\widehat{\mathbf{v}}} \leq K, C > 0)$$

As $G$ and $G'$ were an arbitrary pair of node-level adjacent graph datasets,

$$\Delta_K(\mathbf{u}_t) = \max_{\substack{\deg(G), \, \deg(G') \leq K \\ \text{node-level adjacent} \\ G, G'}} \|\mathbf{u}_t(G) - \mathbf{u}_t(G')\|_F$$

$$< 2(K + 1)C.$$

The proof for the bound on $\Delta_K(\mathbf{u}_t(G))$ when a new node $\widehat{\mathbf{v}}$ is added to the graph $G$ follows analogously. $\qquad \square$

**Lemma 2** (Un-amplified Privacy Guarantee for Each Iteration of Algorithm 1)**.** *Every iteration $t$ of Algorithm 1 is $(\alpha, \gamma)$ node-level Rényi DP for when run on graphs with maximum degree $\leq K$ where:*

$$\gamma = \frac{\alpha \cdot (\Delta_K(\mathbf{u}_t))^2}{2\sigma^2}$$

*Here $\Delta_K(\cdot)$ is the $K$-restricted node-level sensitivity from Definition 3.*

*Proof.* Follows directly from (Mironov, 2017a, Corollary 3). $\qquad \square$

**Lemma 3** (Distribution of Loss Terms Per Minibatch)**.** *For any iteration $t$ in Algorithm 1, consider the minibatch $\mathcal{B}_t$ of subgraphs. For any subset $\mathcal{S}$ of $d$ unique nodes, define the random variable $\rho$ as $|\mathcal{S} \cap \mathcal{B}_t|$. Then, the distribution of $\rho$ follows the hypergeometric distribution Hypergeometric$(N, d, m)$:*

$$\rho_i = P[\rho = i] = \frac{\binom{d}{i}\binom{N-d}{m-i}}{\binom{N}{m}}.$$

*where $N$ is the total number of nodes in the training set $V_{tr}$ and $|\mathcal{B}_t| = m$ is the batch size.*

*Proof.* The minibatches $\mathcal{B}_t$ in Algorithm 1 are formed by sampling nodes from $V_{tr}$ without replacement. When $\rho = i$, one needs to pick $i$ nodes from $\mathcal{S}$ and the remaining $m - i$ nodes from

$V_{tr} - \mathcal{S}$ to form a batch of size $m$. Clearly, there are $\binom{|\mathcal{S}|}{i} = \binom{d}{i}$ ways to do the first step, and $\binom{|S_{tr}|-|\mathcal{S}|}{m-i} = \binom{N-d}{m-i}$ to do the second step. Finally, there are $\binom{N}{m}$ ways to choose a minibatch $\mathcal{B}_t$ of size $m$, each choice equally likely. In conclusion, we can claim:

$$P[\rho = i] = \frac{\binom{d}{i}\binom{N-d}{m-i}}{\binom{N}{m}}$$

which is exactly the Hypergeometric$(N, d, m)$ distribution. $\qquad\square$

**Lemma 4** (Adaptation of Lemma 25 from Feldman et al. (2018))**.** *Let $\mu_0, \ldots, \mu_n$ and $\nu_0, \ldots, \nu_n$ be probability distributions over some domain Z such that:*

$$D_\alpha(\mu_0 \parallel \nu_0) \leq \varepsilon_0$$
$$\ldots$$
$$D_\alpha(\mu_n \parallel \nu_n) \leq \varepsilon_n$$

*for some given $\varepsilon_0, \ldots, \varepsilon_n$.*

*Let $\rho$ be a probability distribution over $[n] = \{0, \ldots, n\}$. Denote by $\mu_\rho$ (respectively, $\nu_\rho$) the probability distribution over Z obtained by sampling $i$ from $\rho$ and then outputting a random sample from $\mu_i$ (respectively, $\nu_i$). Then:*

$$D_\alpha(\mu_\rho \parallel \nu_\rho) \leq \ln \mathbb{E}_{i \sim \rho}\left[e^{\varepsilon_i(\alpha-1)}\right] = \frac{1}{\alpha - 1}\ln \sum_{i=0}^{n} \rho_i e^{\varepsilon_i(\alpha-1)}.$$

*Proof.* Let $\mu'_\rho$ (respectively $\nu'_\rho$) be the probability distribution over $[n] \times Z$ obtained by sampling $i$ from $\rho$ and then sampling a random $x$ from $\mu_i$ (respectively, $\nu_i$) and outputting $(i, x)$. We can obtain $\mu_\rho$ from $\mu'_\rho$ by applying the function that removes the first coordinate; the same function applied gives $\nu_\rho$ from $\nu'_\rho$. Therefore, by the post-processing properties of the Renyi divergence, we obtain that:

$$D_\alpha(\mu_\rho \parallel \nu_\rho) \leq D_\alpha(\mu'_\rho \parallel \nu'_\rho).$$

Now, observe that for every $i \in [n]$ and $x \in Z$, $\mu'_\rho(i, x) = \rho_i \cdot \mu_i(x)$. Therefore,

$$\begin{aligned}
D_\alpha(\mu'_\rho \parallel \nu'_\rho) &= \frac{1}{\alpha - 1}\ln \mathbb{E}_{(i,x) \sim \nu'_\rho}\left[\frac{\mu'_\rho(i,x)}{\nu'_\rho(i,x)}\right]^\alpha \\
&= \frac{1}{\alpha - 1}\ln \mathbb{E}_{i \sim \rho}\left[\mathbb{E}_{x \sim \nu_i}\left[\frac{\mu_i(x)}{\nu_i(x)}\right]^\alpha\right] \\
&= \frac{1}{\alpha - 1}\ln \mathbb{E}_{i \sim \rho}\left[e^{\varepsilon_i(\alpha-1)}\right] \\
&= \frac{1}{\alpha - 1}\ln \sum_{i=1}^{n} \rho_i e^{\varepsilon_i(\alpha-1)}
\end{aligned}$$

as required. $\qquad\square$

**Lemma 5.** *Let $X$ be a non-negative continuous random variable with cumulative distribution function $F_X$ and density $f_X$. Let $g : \mathbb{R}^{\geq 0} \to \mathbb{R}$ be a differentiable function. Then:*

$$\mathbb{E}[g(X)] = g(0) + \int_0^\infty g'(x)(1 - F_X(x))\,dx$$

*Proof.*

$$
\begin{aligned}
\int_0^\infty g'(x)(1 - F_X(x)) \, dx &= \int_0^\infty g'(x) \Pr\left[X > x\right] \, dx \\
&= \int_0^\infty g'(x) \int_x^\infty f_X(t) \, dt \, dx \\
&= \int_0^\infty \int_x^\infty g'(x) f_X(t) \, dt \, dx \\
&= \int_0^\infty \int_0^t g'(x) f_X(t) \, dx \, dt \\
&= \int_0^\infty f_X(t) \left( \int_0^t g'(x) \, dx \right) \, dt \\
&= \int_0^\infty f_X(t) \left( g(t) - g(0) \right) \, dt \\
&= \mathbb{E}[g(X) - g(0)] = \mathbb{E}[g(X)] - g(0).
\end{aligned}
$$

as claimed. $\qquad \square$

An analogous inequality holds for discrete random variables, taking values on $\mathbb{Z}$.

**Lemma 6.** *Let $X$ be a discrete random variable taking values on $\mathbb{Z}$ with cumulative distribution function $F_X$ and probability mass function $f_X$. Let $g : \mathbb{Z} \to \mathbb{R}$ be a function. Then:*

$$
\mathbb{E}[g(X)] = g(0) + \sum_{x=0}^\infty (g(x+1) - g(x))(1 - F_X(x)).
$$

*Proof.* The proof is identical to that of Lemma 5, by replacing integrals with sums. $\qquad \square$

**Lemma 7.** *Let $\rho$ and $\rho'$ be two random variables with the hypergeometric distribution:*

$$
\rho \sim \text{Hypergeometric}(N, k, m)
$$
$$
\rho' \sim \text{Hypergeometric}(N, k', m)
$$

*such that $k \geq k'$. Then, $\rho$ stochastically dominates $\rho'$:*

$$
F_{\rho'}(i) \geq F_\rho(i) \text{ for all } i \in \mathbb{R}
$$

*where $F_\rho$ (respectively, $F_{\rho'}$) is the cumulative distribution function (CDF) of $\rho$ (respectively, $\rho'$).*

*Proof.* Note the following representation of the hypergeometric random variable as the sum of *dependent* Bernoulli random variables:

$$
\rho = \sum_{i=1}^m X_i
$$

where each $X_i \sim \text{Bernoulli}(\frac{k}{N})$. Similarly, we have:

$$
\rho' = \sum_{i=1}^m X_i'
$$

where each $X_i' \sim \text{Bernoulli}(\frac{k'}{N})$. Now, as $k \geq k'$, by a simple analysis for Bernoulli random variables, each $X_i'$ is stochastically dominated by $X_i$:

$$
F_{X_i'} \geq F_{X_i}.
$$

for each $i \in \{1, \dots, m\}$. Thus, as sums preserve stochastic dominance:

$$
F_{\rho'} = F_{\sum_{i=1}^N X_i'} \geq F_{\sum_{i=1}^N X_i} = F_\rho \tag{4}
$$

as required. $\qquad \square$

**Lemma 8.** *Let $\rho$ and $\rho'$ be two non-negative random variables such that $\rho$ stochastically dominates $\rho'$:*

$$F_{\rho'}(i) \geq F_\rho(i) \text{ for all } i \in \mathbb{R}$$

*where $F_\rho$ (respectively, $F_{\rho'}$) is the cumulative distribution function (CDF) of $\rho$ (respectively, $\rho'$).*

*Let $g : \mathbb{R}^{\geq 0} \to \mathbb{R}$ be a non-decreasing differentiable function. Then, the following inequality holds:*

$$\mathbb{E}[g(\rho')] \leq \mathbb{E}[g(\rho)].$$

*Proof.* We first argue for the case where both $\rho$ and $\rho'$ are continuous. By Lemma 5, we have that:

$$\mathbb{E}[g(\rho)] = g(0) + \int_0^\infty g'(x)(1 - F_\rho(x)) \, dx$$

$$\mathbb{E}[g(\rho')] = g(0) + \int_0^\infty g'(x)(1 - F_{\rho'}(x)) \, dx.$$

and hence:

$$\mathbb{E}[g(\rho)] - \mathbb{E}[g(\rho')] = \int_0^\infty g'(x)(F_{\rho'}(x) - F_\rho(x)) \, dx.$$

As $g$ is non-decreasing, we have that $g' \geq 0$ everywhere. The theorem now follows directly.

The case where both $\rho$ and $\rho'$ are discrete can be handled analogously, by using Lemma 6 above instead. $\square$

We are now ready to supply the proof of the main theoretical result in this paper, Theorem 1.

*Proof of Theorem 1.* We borrow notation from the proof of Lemma 1. Let $G$ be an arbitrary graph with adjacency matrix $\mathbf{A}$ and maximum degree bounded above by $K$. Consider an adjacent graph $G'$ with adjacency matrix $\mathbf{A}'$ formed by removing a single node $\widehat{\mathbf{v}}$ from $G$. For convenience, for any node $v$, we denote the corresponding loss terms $\ell_v$ and $\ell'_v$ as:

$$\ell_v = \ell \left( \mathsf{GCN}(\mathbf{A}, \mathbf{X}, v; \boldsymbol{\Theta}); \mathbf{y}_v \right)$$
$$\ell'_v = \ell \left( \mathsf{GCN}(\mathbf{A}', \mathbf{X}', v; \boldsymbol{\Theta}); \mathbf{y}_v \right)$$

As in Lemma 1,

$$\mathbf{u}_t(G) - \mathbf{u}_t(G')$$
$$= \mathrm{Clip}_C(\nabla_{\boldsymbol{\Theta}} \ell_{\widehat{\mathbf{v}}}) \cdot \mathcal{I}[\widehat{\mathbf{v}} \in \mathcal{B}_t] + \sum_{u \in \mathcal{N}_{\widehat{\mathbf{v}}}} \left( \mathrm{Clip}_C(\nabla_{\boldsymbol{\Theta}} \ell_u) - \mathrm{Clip}_C(\nabla_{\boldsymbol{\Theta}} \ell'_u) \right) \cdot \mathcal{I}[u \in \mathcal{B}_t] \qquad (5)$$

where $\mathcal{I}$ is the indicator function. With the notation from Algorithm 1, we have:

$$\tilde{\mathbf{u}}_t(G) = \mathbf{u}_t(G) + \mathcal{N}(0, \sigma^2 \mathbb{I}),$$
$$\tilde{\mathbf{u}}_t(G') = \mathbf{u}_t(G') + \mathcal{N}(0, \sigma^2 \mathbb{I}).$$

We need to show that:

$$D_\alpha(\tilde{\mathbf{u}}_t(G) \,\|\, \tilde{\mathbf{u}}_t(G')) \leq \gamma.$$

Let $\mathcal{S} = \{u \mid u = v \text{ or } u \in \mathcal{N}_{\widehat{\mathbf{v}}}\}$ be the set of nodes 'affected' by the removal of $\widehat{\mathbf{v}}$. From Equation 5, we see that the sensitivity of $\mathbf{u}_t$ depends on the number of nodes in $\mathcal{S}$ that are present in $\mathcal{B}_t$:

$$\|\mathbf{u}_t(G) - \mathbf{u}_t(G')\|_F$$
$$= \left\| \mathrm{Clip}_C(\nabla_{\boldsymbol{\Theta}} \ell_{\widehat{\mathbf{v}}}) \cdot \mathcal{I}[\widehat{\mathbf{v}} \in \mathcal{B}_t] + \sum_{u \in \mathcal{N}_{\widehat{\mathbf{v}}}} \left( \mathrm{Clip}_C(\nabla_{\boldsymbol{\Theta}} \ell_u) - \mathrm{Clip}_C(\nabla_{\boldsymbol{\Theta}} \ell'_u) \right) \cdot \mathcal{I}[u \in \mathcal{B}_t] \right\|_F$$

Let $\rho'$ be the distribution over $\{0, 1, \ldots d_{\widehat{\mathbf{v}}} + 1\}$ of the number of 'affected' nodes in $\mathcal{S}$ present in $\mathcal{B}_t$, that is, $\rho' = |\mathcal{S} \cap \mathcal{B}_t|$. Lemma 3 then gives us that the distribution of $\rho'$ is:

$$\rho' \sim \mathrm{Hypergeometric}(N, d_{\widehat{\mathbf{v}}} + 1, m). \qquad (6)$$

In particular, when $\rho' = i$, exactly $i$ nodes are sampled in $\mathcal{B}_t$. Then, it follows by the same argument in the proof of Lemma 1 that:

$$\Delta_K(\mathbf{u}_t \mid \rho' = i) < 2iC.$$

Thus, conditioning on $\rho' = i$, we see that every iteration is $(\alpha, \gamma_i)$ node-level Rényi DP, by Lemma 2 where:

$$\gamma_i = \alpha \cdot \frac{(2iC)^2}{2\sigma^2} = \alpha \cdot \frac{2i^2C^2}{\sigma^2}. \tag{7}$$

Define the distributions $\mu_i$ and $\nu_i$ for each $i \in \{0, \dots, d_{\widehat{\mathbf{v}}} + 1\}$, as follows:

$$\mu_i = \left[ \tilde{U}(G) \mid \rho' = i \right]$$
$$\nu_i = \left[ \tilde{U}(G') \mid \rho' = i \right]$$

Then, by Equation 7:

$$D_\alpha(\mu_i \parallel \nu_i) \le \gamma_i$$

For the mixture distributions $\mu_{\rho'} = \tilde{U}(G)$ and $\nu_{\rho'} = \tilde{U}(G')$, Lemma 4 now tells us that:

$$\begin{aligned}
D_\alpha(U(G) \parallel U(G')) &= D_\alpha(\mu_{\rho'} \parallel \nu_{\rho'}) \\
&\le \frac{1}{\alpha - 1} \ln \mathbb{E}_{i \sim \rho'} \left[ \exp\left( \gamma_i(\alpha - 1) \right) \right] \\
&= \frac{1}{\alpha - 1} \ln \mathbb{E}_{i \sim \rho'} \left[ \exp\left( \alpha(\alpha - 1) \cdot \frac{2i^2C^2}{\sigma^2} \right) \right] \\
&= \frac{1}{\alpha - 1} \ln \mathbb{E}_{\rho'} \left[ \exp\left( \alpha(\alpha - 1) \cdot \frac{2\rho'^2C^2}{\sigma^2} \right) \right] \\
&= \frac{1}{\alpha - 1} \ln \mathbb{E}\left[ f(\rho') \right].
\end{aligned} \tag{8}$$

where:

$$f(\rho') = \exp\left( \alpha(\alpha - 1) \cdot \frac{2\rho'^2C^2}{\sigma^2} \right)$$

Define another distribution $\rho$ as:

$$\rho \sim \text{Hypergeometric}(N, K + 1, m).$$

As $d_{\widehat{\mathbf{v}}} \le K$, by Lemma 7, $\rho$ stochastically dominates $\rho'$. Then, as $f$ is non-decreasing, Lemma 8 gives us:

$$\mathbb{E}\left[ f(\rho') \right] \le \mathbb{E}\left[ f(\rho) \right]. \tag{9}$$

It follows from Equation 8 and Equation 9 that:

$$D_\alpha(\tilde{U}(G) \parallel \tilde{U}(G')) \le \frac{1}{\alpha - 1} \ln \mathbb{E}_\rho \left[ \exp\left( \alpha(\alpha - 1) \cdot \frac{2\rho^2C^2}{\sigma^2} \right) \right] = \gamma.$$

As this holds for an arbitrary pair of node-level adjacent graphs $G$ and $G'$, we are done. $\qquad \square$

# B    SAMPLING SUBGRAPHS

To bound the sensitivity of the mini-batch gradient in Algorithm 1, we must carefully bound both the in-degree and out-degree of any node in the graph across all training subgraphs. Algorithm 2 outputs a set of training subgraphs that ensures these degree constraints are met.

Note that once the model parameters have been learnt, no such degree restriction is needed at inference time. This means predictions for the 'test' nodes can use the entire neighbourhood information.

---

**Algorithm 2:** Sampling Subgraphs with In-Degree and Out-Degree Constraints

---

**Data:** Graph $G = (V, E, \mathbf{X}, \mathbf{Y})$, Training set $V_{tr}$, Maximum degree $K$.
**Result:** Set of training subgraphs $S_{tr}$.
**for** $v \in V$ **do**
   | Initialize $count_v \leftarrow 0$. Initialize subgraph $S_v \leftarrow \{v\}$.
**end**
Shuffle $V_{tr}$.
**for** $v \in V_{tr}$ **do**
    **for** $u \in \mathcal{N}_v$ **do**
       If $count_u = K$, **continue**.
       If $count_v = K$, **break**.
       Add node $u$ to subgraph $S_v$.
       Add node $v$ to subgraph $S_u$.
       Increment $count_u$ by 1.
       Increment $count_v$ by 1.
    **end**
**end**
Construct $S_{tr} \leftarrow \{S_v \mid v \in V_{tr}\}$.
**return** $S_{tr}$.

---

# C    EXPERIMENTS WITH DIFFERENT GNN ARCHITECTURES

As mentioned in Section 4, the DP-GNN training mechanisms can be used with any 1-layer GNN architecture.

We experiment with different GNN architectures, namely GIN (Xu et al., 2018) and GAT (Veličković et al., 2018) on the ogbn-arxiv dataset and report the results for the respective private and non-private models in Table 4. We use a variant of the original GAT architecture, utilizing dot-product attention instead of additive attention, with 10 attention heads.

We observe that DP-GNN performs reasonably well across different architectures.

Table 4: **Test accuracy of DP-GNN (Adam) on the ogbn-arxiv dataset with a privacy budget of** $\varepsilon \leq 30$.

| Architecture | Non-Private GNN | DP-GNN |
|---|---|---|
| GCN | $68.422 \pm 0.267$ | $63.934 \pm 0.469$ |
| GIN | $67.485 \pm 0.391$ | $63.888 \pm 0.709$ |
| GAT | $65.702 \pm 0.674$ | $58.853 \pm 0.246$ |

# D    LEARNING GRAPH CONVOLUTIONAL NETWORKS (GCN) VIA DP-ADAM

In Algorithm 3, we provide the description of DP-Adam, which adapts Algorithm 1 to use the popular Adam (Kingma & Ba, 2014) optimizer, instead of SGD. The privacy guarantee and accounting for Algorithm 3 is identical to that of Algorithm 1, since the DP clipping and noise addition steps are identical.

---

**Algorithm 3:** DP-GNN (Adam): Differentially Private Graph Neural Network with Adam

---

**Data:** Graph $G = (V, E, \mathbf{X}, \mathbf{Y})$, GNN definition GNN, Training set $V_{tr}$, Loss function $\mathcal{L}$,
    Batch size $m$, Maximum degree $K$, Learning rate $\eta$, Clipping threshold $C$, Noise
    standard deviation $\sigma$, Maximum training iterations $T$, Adam hyperparameters $(\beta_1, \beta_2)$.
**Result:** GNN parameters $\mathbf{\Theta}_T$.
Note that $V_{tr}$ is the subset of nodes for which labels are available (see Paragraph 1 of Section 3).
Using $V_{tr}$, construct the set of training subgraphs $S_{tr}$ with Algorithm 2.
Construct the $0-1$ adjacency matrix $\mathbf{A}$: $\mathbf{A}_{vu} = 1 \iff (v, u) \in S_{tr}$
Initialize $\mathbf{\Theta}_0$ randomly.
**for** $t = 0$ **to** $T$ **do**
    Sample set $\mathcal{B}_t \subseteq V_{tr}$ of size $m$ uniformly at random from all subsets of $V_{tr}$
    Compute the gradient term $\mathbf{u}_t$ as the sum of the clipped gradient terms in the batch $\mathcal{B}_t$:

$$\mathbf{u}_t \leftarrow \sum_{v \in \mathcal{B}_t} \text{Clip}_C(\nabla_{\mathbf{\Theta}} \ell\left(\text{GNN}(\mathbf{A}, \mathbf{X}, v; \mathbf{\Theta}_t); \mathbf{y}_v\right))$$

    Add independent Gaussian noise to the gradient term: $\tilde{\mathbf{u}}_t \leftarrow \mathbf{u}_t + \mathcal{N}(0, \sigma^2 \mathbb{I})$
    Update first and second moment estimators with the noisy gradient, correcting for bias:

$$f_t \leftarrow \beta_1 \cdot f_{t-1} + (1 - \beta_1) \cdot \tilde{\mathbf{u}}_t$$
$$s_t \leftarrow \beta_2 \cdot s_{t-1} + (1 - \beta_2) \cdot (\tilde{\mathbf{u}}_t \odot \tilde{\mathbf{u}}_t)$$
$$\widehat{f_t} \leftarrow \frac{f_t}{1 - \beta_1^t}$$
$$\widehat{s_t} \leftarrow \frac{s_t}{1 - \beta_2^t}$$

    Update the current estimate of the parameters with the noisy estimators:

$$\mathbf{\Theta}_{t+1} \leftarrow \mathbf{\Theta}_t - \frac{\eta}{m} \frac{\widehat{f_t}}{\sqrt{\widehat{s_t}^2} + \varepsilon}$$

**end**

---

# E    EXPERIMENTAL DETAILS AND REPRODUCIBILITY

Table 5 provides details on the benchmark node classification datasets from the OGB suite used in the experiments. The following 3 datasets were used to demonstrate the effectiveness of our method: ogbn-arxiv[6] and ogbn-mag[7] dataset consisting of papers extracted from the Microsoft Academic Graph (MAG) dataset (Wang et al., 2020) and ogbn-products[8] dataset which is a co-purchasing network of Amazon products.

**Hyperparameter configurations for all methods**: We use the following 'inverse-degree' normalization of the adjacency matrix for all GCN models:

$$\widehat{\mathbf{A}} = (d + \mathbb{I})^{-1}(\mathbf{A} + \mathbb{I}).$$

Adam (Kingma & Ba, 2014) with $\beta_1 = 0.9$ and $\beta_2 = 0.999$, and SGD optimizers were used for training all methods for each of the datasets. We fix $C_\%$ as 75.

---

[6] https://ogb.stanford.edu/docs/nodeprop/#ogbn-arxiv    [7] https://ogb.stanford.edu/docs/nodeprop/#ogbn-mag
[8] https://ogb.stanford.edu/docs/nodeprop/#ogbn-products

Table 5: **Statistics of datasets used in our experiments.** On all of these datasets, the task is to classify individual nodes into one of multiple classes.

| Dataset | # Nodes | Avg. Degree | # Features | # Classes | Train/Val/Test Splits |
|---------|---------|-------------|------------|-----------|----------------------|
| ogbn-arxiv | 169,343 | 13.7 | 128 | 40 | 0.54/0.18/0.28 |
| ogbn-products | 61,859,140 | 50.5 | 100 | 47 | 0.08/0.02/0.90 |
| ogbn-mag | 736,389 | 21.7 | 128 | 349 | 0.85/0.09/0.05 |

A dataset-specific grid search was performed over the other hyperparameters for each method, mentioned below. $lr$ refers to the learning rate, $n_{enc}$ refers to the number of layers in the encoder MLP, $n_{dec}$ refers to the number of layers in the decoder MLP, $\lambda$ refers to the noise multiplier, $C_f$ refers to the clipping scaling factor, and $K$ refers to the sampling degree.

**ogbn-arxiv:**

- **Non-Private GCN**: $lr$ in $\{0.001, 0.002, 0.003\}$, $n_{enc}$ in $\{1, 2\}$, $n_{dec}$ in $\{1, 2\}$, Batch Size in $\{1000\}$, Activation in $\{$ReLU$\}$, $K$ in $\{7, 10\}$.
- **DP-GNN**: $lr$ (Adam) in $\{0.001, 0.002, 0.003\}$, $lr$ (SGD) in $\{0.2, 0.5, 1.0\}$, $n_{enc}$ in $\{1, 2\}$, $n_{dec}$ in $\{1, 2\}$, Batch Size in $\{10000\}$, Activation in $\{$Tanh$\}$, $\lambda$ in $\{1.0\}$, $C_f$ in $\{1.0\}$, $K$ in $\{7, 10\}$.
- **Non-Private MLP**: $lr$ in $\{0.001, 0.002, 0.003\}$, $n_{enc}$ in $\{1, 2\}$, $n_{dec}$ in $\{1, 2\}$, Batch Size in $\{1000\}$, Activation in $\{$ReLU$\}$.
- **DP-MLP**: $lr$ in $\{0.001, 0.002, 0.003\}$, $n_{enc}$ in $\{1, 2\}$, $n_{dec}$ in $\{1, 2\}$, Batch Size in $\{10000\}$, Activation in $\{$Tanh$\}$, $\lambda$ in $\{1.0\}$, $C_f$ in $\{1.0\}$.

**ogbn-products:**

- **Non-Private GCN**: $lr$ in $\{0.001, 0.002, 0.003\}$, $n_{enc}$ in $\{1, 2\}$, $n_{dec}$ in $\{1, 2\}$, Batch Size in $\{1000, 4096\}$, Activation in $\{$ReLU, Tanh$\}$, $K$ in $\{10\}$.
- **DP-GNN**: $lr$ (Adam) in $\{0.001, 0.002, 0.003\}$, $lr$ (SGD) in $\{0.01, 0.1, 1.0\}$, $n_{enc}$ in $\{1, 2\}$, $n_{dec}$ in $\{1, 2\}$, Batch Size in $\{1000, 4096, 10000\}$, Activation in $\{$ReLU, Tanh$\}$, $\lambda$ in $\{0.8, 0.9, 1.0\}$, $C_f$ in $\{1.0\}$, $K$ in $\{10\}$.
- **Non-Private MLP**: $lr$ in $\{0.001, 0.002, 0.003\}$, $n_{enc}$ in $\{1, 2\}$, $n_{dec}$ in $\{1, 2\}$, Batch Size in $\{1000, 4096, 10000\}$, Activation in $\{$ReLU, Tanh$\}$.
- **DP-MLP**: $lr$ in $\{0.001, 0.002, 0.003\}$, $n_{enc}$ in $\{1, 2\}$, $n_{dec}$ in $\{1, 2\}$, Batch Size in $\{1000, 4096, 10000\}$, Activation in $\{$ReLU, Tanh$\}$, $\lambda$ in $\{0.8, 0.9, 1.0\}$, $C_f$ in $\{1.0\}$.

**ogbn-mag:**

- **Non-Private GCN**: $lr$ in $\{0.001, 0.002, 0.003\}$, $n_{enc}$ in $\{1, 2\}$, $n_{dec}$ in $\{1, 2\}$, Batch Size in $\{1000, 4096, 5000, 10000\}$, Activation in $\{$ReLU, Tanh$\}$, $K$ in $\{3, 5, 10\}$.
- **DP-GNN**: $lr$ (Adam) in $\{0.001, 0.003, 0.01\}$, $lr$ (SGD) in $\{0.1, 0.5, 0.8, 1.0\}$, $n_{enc}$ in $\{1, 2\}$, $n_{dec}$ in $\{1, 2\}$, Batch Size in $\{1000, 4096, 5000, 10000\}$, Activation in $\{$ReLU, Tanh$\}$, $\lambda$ in $\{1.0, 0.8, 0.5\}$, $C_f$ in $\{1.0, 2.0, 4.0\}$, $K$ in $\{3, 5, 10\}$.
- **Non-Private MLP**: $lr$ in $\{0.001, 0.003, 0.01\}$, $n_{enc}$ in $\{1, 2\}$, $n_{dec}$ in $\{1, 2\}$, Batch Size in $\{1000, 4096, 5000, 10000\}$, Activation in $\{$ReLU, Tanh$\}$.
- **DP-MLP**: $lr$ in $\{0.001, 0.003, 0.01\}$, $n_{enc}$ in $\{1, 2\}$, $n_{dec}$ in $\{1, 2\}$, Batch Size in $\{1000, 4096, 5000, 10000\}$, Activation in $\{$ReLU, Tanh$\}$, $\lambda$ in $\{1.0, 0.8, 0.5\}$, $C_f$ in $\{1.0, 2.0, 4.0\}$.

Additionally, the best hyperparameters corresponding to each experiment to reproduce the results in the main paper are reported in Table 6.

Table 6: Best hyperparameters corresponding to each method across datasets to reproduce the results in the main paper.

| Method | Dataset | $lr$ | Batch Size | Activation | $n_{enc}$ | $n_{dec}$ | $\lambda$ | $C_f$ | $K$ |
|---|---|---|---|---|---|---|---|---|---|
| Non - Private GCN | ogbn-arxiv | 0.001 | 1000 | ReLU | 2 | 1 | | | 10 |
| | ogbn-products | 0.001 | 1000 | ReLU | 2 | 1 | | | 10 |
| | ogbn-mag | 0.001 | 5000 | ReLU | 1 | 1 | | | 5 |
| DP-GNN (Adam) | ogbn-arxiv | 0.003 | 10000 | Tanh | 1 | 2 | 1.0 | 1.0 | 7 |
| | ogbn-products | 0.003 | 10000 | ReLU | 1 | 1 | 0.8 | 1.0 | 10 |
| | ogbn-mag | 0.001 | 4096 | Tanh | 1 | 2 | 1.0 | 4.0 | 5 |
| DP-GNN (SGD) | ogbn-arxiv | 1.0 | 10000 | Tanh | 1 | 1 | 1.0 | 1.0 | 7 |
| | ogbn-products | 1.0 | 10000 | ReLU | 1 | 1 | 0.8 | 1.0 | 10 |
| | ogbn-mag | 1.0 | 10000 | ReLU | 1 | 2 | 1.0 | 1.0 | 5 |
| Non - Private MLP | ogbn-arxiv | 0.001 | 1000 | ReLU | 2 | 1 | | | |
| | ogbn-products | 0.001 | 1000 | ReLU | 1 | 2 | | | |
| | ogbn-mag | 0.01 | 1000 | Tanh | 2 | 2 | | | |
| Private MLP | ogbn-arxiv | 0.003 | 10000 | Tanh | 1 | 2 | 1.0 | 1.0 | |
| | ogbn-products | 0.002 | 10000 | ReLU | 1 | 2 | 0.8 | 1.0 | |
| | ogbn-mag | 0.001 | 10000 | ReLU | 2 | 2 | 1.0 | 1.0 | |

# F    CLASS-WISE ANALYSIS OF LEARNT MODELS

To better understand the performance of the private model as compared to the non-private baseline for our considering setting of multi-class classification at a node-level, we compare the accuracy of these two models for each dataset at a class-wise granularity. These results are summarized in Figure 3. We empirically observe that the performance of the private model degrades as the frequency of training data points for a particular class decreases. This indicates that the model is able to classify data points of "frequent" classes with reasonable accuracy, but struggles with classification accuracy on the data points of "rarer" classes. This observation is in line with previous claims from (Bagdasaryan et al., 2019; Fioretto et al., 2021) that differentially-private models generally perform disparately worse on under-represented classes.

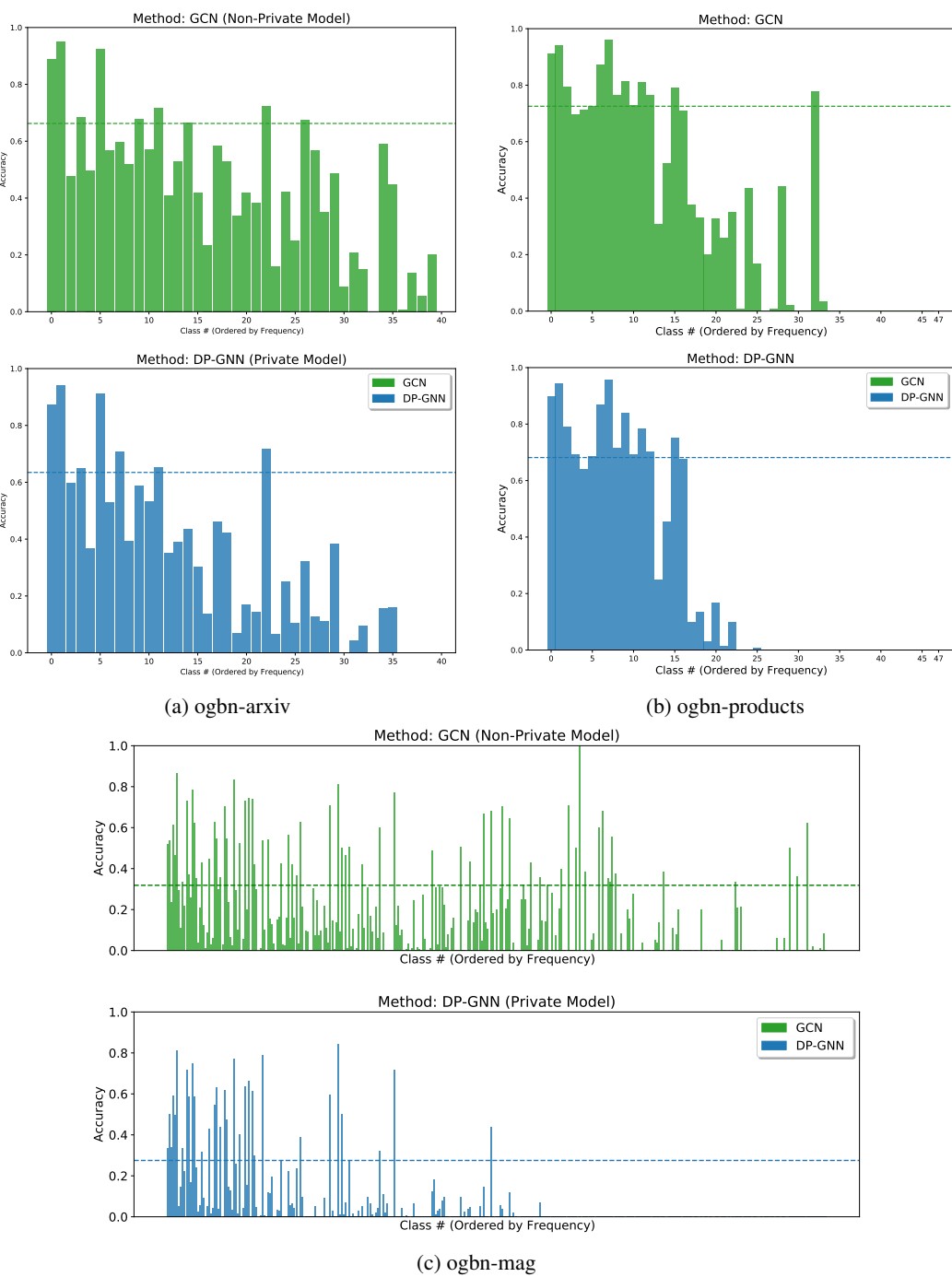

Figure 3: **Comparison of class-wise test accuracies of the non-private GCN model and private DP-GCN model on all datasets, ordered by the decreasing frequency of occurrence of classes in the training data from left to right.** The dotted lines indicate the overall accuracy for the corresponding models. We observe that the private model performs relatively better for classes which have a high frequency in the training data, and the performance degrades as the frequency of the classes decreases.

