# OpenReview forum: "Node-Level Differentially Private Graph Neural Networks"
_ICLR.cc/2022/Conference — ICLR 2022 Submitted_

### Official Review · Reviewer_Xcpu · 2021-10-24

**Correctness:** 3
**Technical Novelty And Significance:** 2
**Empirical Novelty And Significance:** 1
**Recommendation:** 3
**Confidence:** 3

**Main Review:**

## Strong pints
1. The main motivation of the paper is valid and active research area.
2. It was written well and reasonable clear.

## Weak Points
Even though I have no experiences on private networks, I did some reading on it but try to judge the paper on mostly point of GNN view.
Here is the points that I wanted to discuss with the authors.

1. Differentially private networks protect training data from the end user. In this submission it is not clear from whom we need to protect node features and adjacency. Sometimes I thought they tried to protect node features and adjacency of each node from all rest of the nodes, but not sure. Thus, the usage scenario should be given by a toy example. For instance in (Wu et al., 2021b) there is great example scenario where Bob is customer  has node features and Alice is ML developer has adjacency. Bob sends this data to Alice to develop a GNN for some specific tasks. Alice should provide a trained model where Bob cannot find any part of adjacency by querying to trained model. So their aim is to provide edge level data privacy. Here the aim is to provide privacy both node feature and edge information, it seems the node features and adjacency should not be known by other nodes. But what about server side if the model is centralize? Do the central model knows all raw train data?

2. In Related Work section, Graph level prediction tasks was cited as they are in different contexts. However, graph based prediction or node level predictions are not so different to each other. Any GNN architecture can be used for either node (or edge) or graph level prediction. Just the differences is to apply graph level readout after last GNN layer. So, whatever exist in graph level task while preserving privacy should be direct competitor of the proposed method.

3. Node based prediction under transductive problem settings is the easiest problem setting for GNN. Because there is just one graph and  the graph is known in advance. Transferabililty from train graph to the test graph is not the problem. As it is mentioned example scenario in Introduction, the method must be tested under inductive problem settings, where the GNN learns something from train graph ( in example large enterprise) and tested on test graph ( small enterprise). In this way we can see if the learned coefficient can be transferable to the another graph well or not.

4. The authors main concentration was on node level tasks. However there are much more important tasks on graph level prediction while node level privacy is still important (as the same scenario in Introduction, train graphs would belongs some enterprises and task would be predict something about each enterprise. Then we can test given new enterprise's graph by predicting enterprises level output).  The experiments can be extended on this context.

5. Although  (Wu et al., 2021a)'s proposition was tested on bipartite graphs (user to items graphs), since they used a general GNN (the GNN architecture was not changed for bipartite graphs), it can easily extendable on concerned dataset in the paper. Thus their proposal has strong connection to this paper. I think the differences should be discussed in the paper and their method should be used as another baseline.

6. Eq1 and Eq2 are not the general formulation of GNN. Usually we show it by message aggregation and update functions.  More specifically, the author defines GNN as arbitrary number of MLP on node features, aggregation operation based on graph adjacency, followed by arbitrary number of MLP on aggregated features. They used exactly one layer of aggregation but one or two MLP for both on node features and aggregated features.  For sure this equation defines a GNN, but not definitely GCN (Kipf 2016) GAT (velickovic, 2018) and other mentioned models (GIN, Xu 2018, GraphSage Hamilton 2017).

7. Seems in the literature, differentially private networks results were performed for different privacy costs(Abadi et al., 2016). In this submission, it is selected as \epsilon=30. It had better if we can see the result under different privacy cost in 2D plot, like Cost-accuracy scatter plot. I would like to know at what privacy cost, the accuracy of proposed method will be the same with non-private GNN.

8. Is there any specific reason to do analyse for just single GNN layer? Is it extendable for arbitrary number of GNN layers?

9. Does any specific GNN methods (GCN, GAT, ChebNet, GIN ...) perform better than others? In the paper just one type of special GNN (given in Eq1-2) is used. Can any well-known GNN be used in this context?











**Summary Of The Paper:**

The paper claim to proposed a novel training method of GNN under providing both node feature and adjacency privacy. Their main approach is to use existed differential privacy framework into GNN world. They claim that providing privacy on both node feature and their connectivity is novel.

**Summary Of The Review:**

Proposed method seems an application of existed DP method for the graph data. So the novelty of the paper is limited. Experimental section is insufficient. I recommend rejection for this paper.

---

> ### Author Response · Authors · 2021-11-17
> **Response to Reviewer Xcpu (part 1)**
>
> We would like to thank the reviewer for their time and constructive feedback for our paper.
>
> Since our original submission, we have identified an improvement in the sampling scheme. In particular, at training time, node neighborhoods need to be sampled carefully to form subgraphs. However, at inference time, once the GNN parameters have been learnt, the entire neighborhood can be used as input to the GNN. The new sampling scheme is described in Appendix B. This scheme has given a performance improvement on all 3 OGB datasets, with performance outperforming MLP and DP-MLP models even at epsilon <= 15. We have updated our results, tables and plots accordingly.
>
> Further, to demonstrate the general applicability of our DP-GNN method, we have added results on ogbn-arxiv with the GIN and GAT architectures.
>
> Below, we have addressed specific points from the review.
>
> > **But what about server side if the model is centralize? Do the central model knows all raw train data?**
>
> The centralized setting for learning private GNNs is not new, see (Wu et al., 2021a at https://arxiv.org/abs/2102.04925) for example. Similar to their setting, the central ‘server’ is trusted only with the individual gradient terms to compute the mini-batch gradient in a differentially private manner.
>
> > **So, whatever exist in graph level task while preserving privacy should be direct competitor of the proposed method.**
>
> We would like to point out that there do not exist any node-level differentially private GNNs for graph-level prediction either. As the reviewer points out, this is because the design of modern GNNs consists of a readout function applied to node-level predictions. Thus, to create a node-level differentially-private GNN to solve graph-level tasks, one must first create a node-level differentially-private GNN that can generate node-level predictions, as we are the first to do here.
>
> > **Node based prediction under transductive problem settings is the easiest problem setting for GNN.**
>
> This may be true in the non-private setting, but to solve transductive node-level prediction problems under the constraint of node-level differential privacy is challenging, as we have explained and demonstrated here.
>
> > **Although (Wu et al., 2021a)'s proposition was tested on bipartite graphs (user to items graphs), since they used a general GNN (the GNN architecture was not changed for bipartite graphs), it can easily extendable on concerned dataset in the paper.**
>
> We disagree with the reviewer’s conclusion here. While the GNN architecture from (Wu et al., 2021a) might be successfully extended to general graphs, the other essential parts of their algorithm (such as Privacy-Preserving User-Item Graph Expansion) are only probably private when dealing with bipartite graphs. This is why (Wu et al., 2021a) focuses on the specific problem of user-item recommendation systems, since the problem naturally lends itself to a bipartite graph structure.
>
> > **For sure this equation defines a GNN, but not definitely GCN (Kipf 2016), GAT (Velickovic, 2018) and other mentioned models (GIN, Xu 2018, GraphSage Hamilton 2017).**
>
> This equation does define a GCN model. In fact, our original submission had experiments only for the GCN model. We have now added results for the GAT and GIN models on the ogbn-arxiv dataset as well, in Appendix C.
>
> > **It had better if we can see the result under different privacy cost in 2D plot, like Cost-accuracy scatter plot.**
>
> Figure 1 is precisely this: a comparison of the privacy cost vs the test accuracy.
>
> > **I would like to know at what privacy cost, the accuracy of proposed method will be the same with non-private GNN.**
>
> This is an interesting question. In our experiments we found that the performance of the DP-GNN models saturates beyond a point, not reaching the non-private GNN performance. This behavior under DP-SGD training has been seen in previous research as well (https://arxiv.org/abs/2007.14191) and can be attributed to the bias to the gradients introduced by clipping and noise addition during training.

---

> > ### Author Response · Authors · 2021-11-17
> > **Response to Reviewer Xcpu (part 2)**
> >
> > (continuing the response above due to space limits)
> >
> > > **Does any specific GNN methods (GCN, GAT, ChebNet, GIN ...) perform better than others? In the paper just one type of special GNN (given in Eq1-2) is used. Can any well-known GNN be used in this context?**
> >
> > Our algorithm (and resulting privacy guarantee) is valid for any 1-layer GNN, such as the ones being mentioned. Our original submission had results with the GCN model only. We have now added results for the GAT and GIN models on the ogbn-arxiv dataset as well, in Appendix C. For convenience, we reproduce the table below. We see that across all three GNN architectures, DP-GNN can outperform MLP baselines.
> >
> > **Table:** Test accuracies of our DP-GNN (Adam) method with different GNN architectures on the ogbn-arxiv dataset, with a privacy budget of $\epsilon \leq 30$
> >
> > | **Algorithm**  |  **Non-Private GNN** | **Private DP-GNN** |
> > | ----------- | ----------- |----------- |
> > | GCN | 68.422 $\pm$ 0.267 | 63.934 $\pm$ 0.469 |
> > | GIN | 67.485 $\pm$ 0.391 | 63.888  $\pm$ 0.709 |
> > | GAT | 65.702 $\pm$ 0.674 | 58.853 $\pm$ 0.246 |
> >
> >
> > > **Proposed method seems an application of existed DP method for the graph data. So the novelty of the paper is limited.**
> >
> > As we have described in Section 4, existing DP-SGD analysis mostly applies to “edge level” privacy. But in several practical settings, each node can represent a user, so providing node-level privacy is critical in several tasks. Node-level DP is challenging because the removal of a single node affects multiple gradient terms. This is why we perform a careful sensitivity analysis and come up with a novel ‘privacy amplification by subsampling’ theorem that generalizes previously known theorems. This ensures the successful application of DP-SGD for training GNNs in a node-level differentially private manner.

---

> ### Author Response · Authors · 2021-11-19
> **Follow-up on response to Reviewer Xcpu**
>
> Dear Reviewer Xcpu,
>
> We would like to thank you for your insightful comments and suggestions. We hope that our responses and revision address your concerns satisfactorily and will lead to a more positive evaluation of our work. With the discussion window closing soon, we wanted to know if there are any further clarifications that we can provide from our end.

---

### Official Review · Reviewer_upoR · 2021-11-02

**Correctness:** 1
**Technical Novelty And Significance:** 2
**Empirical Novelty And Significance:** 2
**Recommendation:** 5
**Confidence:** 4

**Main Review:**

Strengths:
1.	It is the first work to provide strong privacy guarantees for each individual node in graph learning, and designs an optimization algorithm to achieve that;
2.	It theoretically proves that the proposed algorithm is guaranteed to be differential private;
3.	The paper is well-structured and easy to follow.

Weakness:
1. This work proves the differential privacy only on 1-layer GNNs. The methodology is directly extended from (Abadi et al., (2015)) and Feldman et al. (2018), with the consideration that a subset of K neighbors could appear in each update step. However, usually two or more layers are used for the graph-related tasks. Showing the form of DP in the more general r-layer case would make the contribution more significant.

2. The experiments are not complete. Why other DP GNN approaches are not compared with? It only compared with vanilla GCN and MLP.

3. The introduction is a little unclear. It talks about “preserves the privacy of the features of each node (‘user’), their labels as well as their connectivity information”, which is a little vague and no examples are provided to show the kind of privacy it can preserve. From the methodology and proof part, it seems to be that whether the graph containing a certain node cannot be detected. As a result, the motivation and application of preserving this type of privacy is not well-introduced.

Besides, the introducing of background in introduction seems to be incorrect.  It introduced (Zhou et al., 2020b) as an existing work on “bi-partite graphs or node-level privacy without preserving individual connectivity”, but that work does not focus on graph at all.

A typo seems to exist in proof of Lemma 7, should the ρ’ in first equation be changed to ρ?

**Summary Of The Paper:**

This paper aims to achieve node-level differential privacy on GNNs. Through adding noises to calculated gradients at each optimization step, this work shows that trained GNN layer can be (α, γ) node-level Renyi differentially-private. Concretely, this work considers the 1-layer GNN case, and make the maximum degree of each node to be K. With these requirements, this work theoretically prove the scale of noises.

Its contributions are as follows:
1.	Propose the task of learning node-level differential private GNNs,
2.	Adapt DP-SGD to work on graphs, through extending amplified privacy guarantee,
3.	Evaluate proposed optimization algorithm on benchmark graph datasets.

**Summary Of The Review:**

This paper considers the node-level DP problem on graphs, which is the first work on this task as far as I can see.  The problem is formally defined, and guaranteed privacy is proved.

The methodology is directly extended from (Abadi et al., (2015)) and Feldman et al. (2018), with the consideration that a subset of K neighbors could appear in each update step. However, the proof is incomplete as only 1-layer GNN is considered. Furthermore, none of existing DP GNNs are adopted as baseline approaches.

---

> ### Author Response · Authors · 2021-11-17
> **Response to Reviewer upoR**
>
> We would like to thank the reviewer for their time and constructive feedback for our paper.
>
> Since our original submission, we have identified an improvement in the sampling scheme. In particular, at training time, node neighborhoods need to be sampled carefully to form subgraphs. However, at inference time, once the GNN parameters have been learnt, the entire neighborhood can be used as input to the GNN. The new sampling scheme is described in Appendix B. This scheme has given a performance improvement on all 3 OGB datasets, with performance outperforming MLP and DP-MLP models even at epsilon <= 15. We have updated our results, tables and plots accordingly.
>
> Further, to demonstrate the general applicability of our DP-GNN method, we have added results on ogbn-arxiv with the GIN and GAT architectures.
>
> Below, we have addressed specific points from the review.
>
> > **The methodology is directly extended from (Abadi et al., (2015)) and Feldman et al. (2018), with the consideration that a subset of K neighbors could appear in each update step.**
>
> We would like to point out to the reviewer that the extension from existing DP-SGD analyses is non-trivial, because of the fact that the removal of a single node affects multiple subgraphs, which can occur together in a single mini-batch. Our novel theoretical contribution is to show that a ‘privacy amplification by subsampling’ theorem still holds in this case.
>
> > **Showing the form of DP in the more general r-layer case would make the contribution more significant.**
>
> As mentioned in the introduction, we can easily adapt our existing algorithm to 2 and more layer GNNs. The key issue is the privacy setting when performing node-level prediction. In particular, if using a r-layer GNN, then at inference time for a node u, the features of all other nodes v in the entire r-layer neighborhood of u need to be available non-privately. We felt that this was probably too restrictive in the general case, and have instead focused on 1-layer GNNs. Further, on the datasets explored here, 1-layer GNNs are quite competitive with 2 and more layer GNNs, with the addition of more layers in the MLP encoder and decoder.
>
> > **Why other DP GNN approaches are not compared with?**
>
> Our work is the first to demonstrate the node-level differential private training of GNNs. As explained in the Related Work (Section 2), other approaches either do not preserve node-level privacy, or are created for specific applications (such as recommendation systems) with no direct way to extend to the general node-level prediction setting. Thus, any comparison to other methods would not be justified.
>
> > **The introduction is a little unclear. It talks about “preserves the privacy of the features of each node (‘user’), their labels as well as their connectivity information”, which is a little vague and no examples are provided to show the kind of privacy it can preserve. From the methodology and proof part, it seems to be that whether the graph containing a certain node cannot be detected. As a result, the motivation and application of preserving this type of privacy is not well-introduced.**
>
> The goal of our paper is to enable the training of GNNs in a node-level differentially private manner. Roughly, what this means is the learned GNN parameters should not overly depend on any single node. If a node (its features, labels and connectivity information) is removed from the graph, the GNN parameters should not change significantly. The notion of node-level differentially private graph algorithms has been explored extensively in previous works as well (https://pennstate.pure.elsevier.com/en/publications/private-analysis-of-graph-structure and https://arxiv.org/abs/1504.07912, for example).
>
> > **Besides, the introducing of background in introduction seems to be incorrect. It introduced (Zhou et al., 2020b) as an existing work on “bi-partite graphs or node-level privacy without preserving individual connectivity”, but that work does not focus on graph at all.**
>
> Apologies, and thank you for catching this. We meant to cite Zhou et al., 2020a instead, as mentioned in the Related Work section. This has now been fixed in our latest revision.
>
> > **A typo seems to exist in proof of Lemma 7, should the ρ’ in first equation be changed to ρ?**
>
> Apologies, and thank you for catching this. This has now been fixed in our latest revision.

---

> ### Author Response · Authors · 2021-11-19
> **Follow-up on response to Reviewer upoR**
>
> Dear Reviewer upoR,
>
> We would like to thank you for your insightful comments and suggestions. We hope that our responses and revision address your concerns satisfactorily and will lead to a more positive evaluation of our work. With the discussion window closing soon, we wanted to know if there are any further clarifications that we can provide from our end.

---

### Official Review · Reviewer_Uu9v · 2021-11-02

**Correctness:** 4
**Technical Novelty And Significance:** 2
**Empirical Novelty And Significance:** 2
**Recommendation:** 3
**Confidence:** 3

**Main Review:**


The stated privacy parameters of 30 seems high, although there have been other industrial use cases with such parameters.  I would be interested to know whether edge-level privacy with significantly lower privacy parameters actually ensures more privacy, in some sense, than node-level privacy with such a high parameter.  For edge-level, one can make statements about how if a node has a bounded number of edges, then one can make a node-level privacy guarantee by multiplying the privacy parameters accordingly.  It may not always be the case that you can go the other way, where node-level privacy of 30 gives a much lower edge privacy guarantee.

The privacy guarantee does not seem surprising because if any node is guaranteed to have at most K neighbors, then the node’s influence expands to K other nodes.  That is, to ensure edge level privacy, constant noise would ensure constant privacy parameters, but considering addition or removal of K edges, adding noise proportional to K would ensure constant privacy parameters.  Is the analysis presented here some how improving over this intuition or is there a more technical reason why this intuition breaks down?  This should be made more explicit.

For the experiments section, the gradient clipping procedure depends on the data, in particular the scaling factor.  The scaling factor is fine-tuned on each dataset, so is privacy really ensured?  Furthermore, an exhaustive grid search is done over several hyper-parameters.  In Figure 1, it looks like DP-MLP does better than DP-GNN for moderate privacy parameters, say < 7.  Is there any reason for why this might be and at what point would one method do better than another?

Although the proposed procedure seems to improve over existing approaches for some privacy regimes, I would have liked to see more analysis on when the improvement actually occurs (when epsilon > some value).  Furthermore, there are several parameters that need to be tuned, so it is not clear if the extra privacy budget expended for choosing these parameters would actually lead to a better privacy guarantee than the edge-level privacy.  I also do not see the novelty in the algorithm, since we could just pre-process each node to have at most K edges then run an edge-level private algorithm to get the same node-level privacy guarantee.

### UPDATE ###
The paper could significantly benefit from the feedback given here and the comments they plan to make.  I am still not convinced of the novelty here in the privacy analysis.  The authors point out that "standard composition results do not allow for multiple changes in the input dataset" but DP ensures group level privacy, so in particular can ensure k*eps-DP for groups of size k.  I will keep my score unchanged.


**Summary Of The Paper:**

This work studies the problem of ensuring node-level privacy when training GNNs.  This privacy setting is much stronger than edge-level privacy because it considers the impact a single node can have on training, which may be connected to several other nodes, whereas edge-level privacy considers only the impact of a single edge in the graph.


**Summary Of The Review:**

Although the proposed procedure seems to improve over existing approaches for some privacy regimes, I would have liked to see more analysis on when the improvement actually occurs (when epsilon > some value).  Furthermore, there are several parameters that need to be tuned, so it is not clear if the extra privacy budget expended for choosing these parameters would actually lead to a better privacy guarantee than the edge-level privacy.  I also do not see the novelty in the algorithm, since we could just pre-process each node to have at most K edges then run an edge-level private algorithm to get the same node-level privacy guarantee.

---

> ### Author Response · Authors · 2021-11-17
> **Response to Reviewer Uu9v**
>
> We would like to thank the reviewer for their time and constructive feedback for our paper.
>
> Since our original submission, we have identified an improvement in the sampling scheme. In particular, at training time, node neighborhoods need to be sampled carefully to form subgraphs. However, at inference time, once the GNN parameters have been learnt, the entire neighborhood can be used as input to the GNN. The new sampling scheme is described in Appendix B. This scheme has given a performance improvement on all 3 OGB datasets, with performance outperforming MLP and DP-MLP models even at epsilon <= 15. We have updated our results, tables and plots accordingly.
>
> Further, to demonstrate the general applicability of our DP-GNN method, we have added results on ogbn-arxiv with the GIN and GAT architectures.
>
> Below, we have addressed specific points from the review.
>
> > **The stated privacy parameters of 30 seems high, although there have been other industrial use cases with such parameters.**
>
> As the problem is more difficult than the standard supervised learning (from a privacy point of view), we used epsilon <= 30 as the privacy budget. But, we can train our models for different privacy budgets (such as epsilon <= 10). As mentioned above, in our revised results, we can train DP-GNN models that outperform both MLP and DP-MLP methods at epsilon <= 10 on ogbn-arxiv and ogbn-products, and epsilon <= 15 on ogbn-mag.
>
> > **That is, to ensure edge level privacy, constant noise would ensure constant privacy parameters, but considering addition or removal of K edges, adding noise proportional to K would ensure constant privacy parameters. Is the analysis presented here some how improving over this intuition or is there a more technical reason why this intuition breaks down?**
>
> The reviewer is correct in this preliminary analysis. However, one must be careful as one cannot directly compose edge-level privacy mechanisms to create node-level differential privacy, as standard composition results do not allow for multiple changes in the input dataset. Indeed, our key contribution is to show that the analogous ‘privacy amplification by subsampling’ theorem holds when training GNNs. To show this, we have performed a careful analysis of the sampling scheme and the sensitivity of the mini-batch gradient. Such a result cannot be obtained from naively composing edge-level differentially private mechanisms.
> Even intuitively, it’s not clear if an edge-level argument would work. Since the GNN parameters are shared across all nodes, one node’s embedding can depend on *any* edge in the graph, instead of only the ‘nearby’ K edges mentioned by the reviewer.
>
> > **For the experiments section, the gradient clipping procedure depends on the data, in particular the scaling factor. The scaling factor is fine-tuned on each dataset, so is privacy really ensured?**
>
> As shown in Appendix D, we did not tune the clipping scaling factor for ogbn-arxiv and ogbn-products. Only on ogbn-mag did we explore tuning the clipping scaling factor, giving us an improvement of 1 percentage point (29.5 vs 30.5). In our experience, the clipping scaling factor does not affect the final model performance significantly.
>
> > **Although the proposed procedure seems to improve over existing approaches for some privacy regimes, I would have liked to see more analysis on when the improvement actually occurs (when epsilon > some value).**
>
> In our updated results, we have shown that DP-GNN outperforms DP-MLP over all privacy regimes on both ogbn-arxiv and ogbn-products. On ogbn-mag, the absolute performance gap between the GCN and MLP methods is quite low already. Since the DP-GNN method adds more noise at each step than the DP-MLP (O(K) vs O(1) where K is the sampling degree), the DP-GNN performance is lower for small privacy budgets on ogbn-mag.
>
> > **Furthermore, there are several parameters that need to be tuned, so it is not clear if the extra privacy budget expended for choosing these parameters would actually lead to a better privacy guarantee than the edge-level privacy.**
>
> The only additional hyperparameter for node-level private training is the degree of subsampling, K. The remaining privacy-related hyperparameters (clipping percentile and clipping scaling factor) are present for DP-SGD training in general. The remaining hyperparameters (batch size, learning rate, and so on) are inherited from the base GNN model.  Given the complexity of training large scale non-convex models with DP, and with the lack of any known privacy leakage through hyperparameters, it has become customary to tune the hyperparameters non-privately, and provide the privacy guarantee for the final training run. It is indeed an important question to tune these hyperparameters with privacy. We plan to explore this question in future incarnations of the work, especially in the context of this recent work (https://openreview.net/forum?id=-70L8lpp9DF).

---

> ### Author Response · Authors · 2021-11-19
> **Follow-up on response to Reviewer  Uu9v**
>
> Dear Reviewer Uu9v,
>
> We would like to thank you for your insightful comments and suggestions. We hope that our responses and revision address your concerns satisfactorily and will lead to a more positive evaluation of our work. With the discussion window closing soon, we wanted to know if there are any further clarifications that we can provide from our end.

---

### Official Review · Reviewer_NdP7 · 2021-11-08

**Correctness:** 4
**Technical Novelty And Significance:** 2
**Empirical Novelty And Significance:** 2
**Recommendation:** 6
**Confidence:** 3

**Main Review:**

The extension of the DP-SGD technique to the graph neural networks is very interesting, and the empirical performance seems improved.  I have the below questions.

1. In algorithm 1, it aggregates subgraphs to get a set of subgraphs S_tr. And then it doesn't appear in any of the following steps. I'm confused about how it plays a role in the algorithm?

2. One of the main modifications is that they subsample the neighborhood of each node to ensure that each node has only K neighbors. Then they add noise according to the sensitivity of aggregated gradient wrt an individual node. So basically, I think it works like using group privacy (also, there's a K*C term in lemma1). Somehow, I feel the algorithm just extends the traditional DP-SGD to something like the group privacy case (technically, the privacy is still at the node-level, but it just restricts the case to K corrected nodes). Then in the experimental sections, I feel the benefit over DP-MLP is from incorporating the graph information. Overall, the algorithm is not that very exciting.

3. The paper claims that the mini-batch is uniformly sampled from all training nodes, which contrasts the sampling with replacement in the traditional method. What are the differences? There seems no discussion about why this modification is important for the privacy amplification results.

4. For Table 2 and Table 3, what's the privacy parameter epsilon?




**Summary Of The Paper:**

This paper proposes a private algorithm for Graph Neural Networks at the node level. The algorithm is based on some modifications to the DP-SGD, and it applies to directed graphs. The authors analyze the privacy guarantees through Renyi differential privacy and give amplified privacy guarantees for their algorithm. Empirical evaluation is provided to demonstrate the efficacy of the proposed algorithm.

**Summary Of The Review:**

In summary, I think the technical contribution of this paper is not significant, but it's also worth having a DP algorithm for the GNNs.

---

> ### Author Response · Authors · 2021-11-17
> **Response to Reviewer NdP7**
>
> We would like to thank the reviewer for their time and constructive feedback for our paper.
>
> Since our original submission, we have identified an improvement in the sampling scheme. In particular, at training time, node neighborhoods need to be sampled carefully to form subgraphs. However, at inference time, once the GNN parameters have been learnt, the entire neighborhood can be used as input to the GNN. The new sampling scheme is described in Appendix B. This scheme has given a performance improvement on all 3 OGB datasets, with performance outperforming MLP and DP-MLP models even at epsilon <= 15. We have updated our results, tables and plots accordingly.
>
> Further, to demonstrate the general applicability of our DP-GNN method, we have added results on ogbn-arxiv with the GIN and GAT architectures.
>
> Below, we have addressed specific points from the review.
>
> > **In algorithm 1, it aggregates subgraphs to get a set of subgraphs S_tr. And then it doesn't appear in any of the following steps. I'm confused about how it plays a role in the algorithm?**
>
> The subgraphs S_tr are used by the GNN models to compute the forward and backward passes (hence, the gradient terms). We apologize for missing this earlier and have now clarified this in our latest revision.
>
> > **So basically, I think it works like using group privacy (also, there's a K*C term in lemma1). Somehow, I feel the algorithm just extends the traditional DP-SGD to something like the group privacy case (technically, the privacy is still at the node-level, but it just restricts the case to K corrected nodes).**
>
> The reviewer is correct that such an analysis can be extended to other cases where group privacy needs to be ensured. However, there does not exist a ‘group privacy amplification by subsampling’ theorem that we can use directly. Hence, our novel contribution is to first state and prove a theorem of that form, and then show that it applies to the analysis of training GNNs in a privacy-preserving manner.
>
> > **Overall, the algorithm is not that very exciting.**
>
> We have demonstrated why pre-existing DP-SGD paradigms cannot be applied directly to train GNNs, and have proposed an extension that enables training 1-layer GNNs. Our key contributions are in the novel privacy amplification by subsampling result, and in the successful empirical performance of our DP-GNN models.
>
> > **I feel the benefit over DP-MLP is from incorporating the graph information.**
>
> Indeed, this is the key point of the paper, to show that graph information can be incorporated without sacrificing privacy.
>
> > **The paper claims that the mini-batch is uniformly sampled from all training nodes, which contrasts the sampling with replacement in the traditional method. What are the differences? There seems no discussion about why this modification is important for the privacy amplification results.**
>
> Privacy amplification by sampling has both variants, one with replacement, and one without. The latter is at times cumbersome to use as one might have multiple copies of the same data record in the minibatch, and one has to incorporate that in the sensitivity analysis. This is why we went with the first choice, sampling with replacement. For the GNN case, we must analogously sample subgraphs without replacement, to ensure we can bound the sensitivity correctly and successfully apply the privacy amplification theorem.
>
> > **For Table 2 and Table 3, what's the privacy parameter epsilon?**
>
> The privacy parameter is 30, as everywhere else. We have updated the captions in our latest revision.

---

> ### Author Response · Authors · 2021-11-19
> **Follow-up on response to Reviewer NdP7**
>
> Dear Reviewer NdP7,
>
> We would like to thank you for your insightful comments and suggestions. We hope that our responses and revision address your concerns satisfactorily and will lead to a more positive evaluation of our work. With the discussion window closing soon, we wanted to know if there are any further clarifications that we can provide from our end.

---

### Decision · Program_Chairs · 2022-01-20

**Decision:**

Reject

**Comment:**

This paper received some additional discussion between the reviewers and the area chair. The reviewers were largely unswayed by the author responses. One concern was the level of technical novelty, feeling that this was largely a straightforward adaptation of DPSGD (as, admittedly, most works in the DP ML setting are). The primary technical contribution may be the sampling amplification theorem, which one reviewer felt was also straightforward from previous work. Other criticisms was that the privacy parameter epsilon is rather large, and that results are restricted to 1-layer GNNs. Generally, the work did not feel very novel to reviewers from either the privacy or the GNN community. However, they felt that the paper could benefit substantially from exploration and implementation of the comments made in the responses, so the authors are encouraged to pursue those directions. Some of the many suggestions from reviewer Xcpu may help the authors make the paper appeal more to the GNN community.